# LAZY-CFR: FAST AND NEAR-OPTIMAL REGRET MINIMIZATION FOR EXTENSIVE GAMES WITH IMPERFECT INFORMATION

**Yichi Zhou, Tongzheng Ren, Dong Yan, Jialian Li, Jun Zhu**[*]
Dept. of Comp. Sci. & Tech., BNRist Center, Institute for AI, Tsinghua University; RealAI
`{vofhqn,rtz19970824,sproblvem}@gmail.com,lijialian7@163.com,`
`dcszj@tsinghua.edu.cn`

## ABSTRACT

Counterfactual regret minimization (CFR) methods are effective for solving two-player zero-sum extensive games with imperfect information. However, the vanilla CFR has to traverse the whole game tree in each round, which is time-consuming in large-scale games. In this paper, we present Lazy-CFR, a CFR algorithm that adopts a lazy update strategy to avoid traversing the whole game tree in each round. We prove that the regret of Lazy-CFR is almost the same as the regret of the vanilla CFR and only needs to visit a small portion of the game tree. Thus, Lazy-CFR is provably faster than CFR. Empirical results consistently show that Lazy-CFR is fast in practice.

## 1 INTRODUCTION

Extensive games provide a mathematical framework for modeling the sequential decision-making problems with imperfect information, which is common in economic decisions, negotiations and security. We focus on solving *t*wo-player zero-sum *e*xtensive *g*ames with *i*mperfect information (TEGI). In a TEGI, there is an environment with uncertainty and two players on opposite sides (Koller & Megiddo, 1992).

Counterfactual regret minimization (CFR) (Zinkevich et al., 2008) provides a state-of-the-art approach for solving TEGIs with much progress in practice (Brown & Sandholm, 2017b; Moravčík et al., 2017; Brown & Sandholm, 2019a). Regret minimization techniques are first introduced to solve TEGIs based on the observation that minimizing the regrets of both players makes the time-averaged strategy converge to the Nash Equilibrium (NE) (Nisan et al., 2007). CFR (Zinkevich et al., 2008) further bounds the original regret with a summation of many immediate counterfactual regrets on each information set (infoset). These immediate counterfactual regrets are defined by the counterfactual rewards and can be iteratively minimized by existing online learning algorithms, e.g., regret matching (RM) (Blackwell et al., 1956).

A limitation of CFR is that it requires traversing the whole game tree in each round, which is time-consuming in large-scale games due to the fact that we have to apply RM to every immediate regret in each round. Existing works on avoiding traversing the whole game tree can be mainly divided into two categories: Pruning-based CFR (Brown & Sandholm, 2015; 2017a) and Monte-Carlo CFR (MC-CFR) (Lanctot et al., 2009). These algorithms can significantly speed up the vanilla CFR in practice. However, pruning-based algorithms may degenerate in the worst case. And the performance of MC-CFR depends on the structure of the game and the chosen online learning algorithm.

**Contributions:** In this paper, we explore another approach to address the problem.

- We present an algorithm named Lazy-CFR, which exploits a lazy update technique to avoid traversing the whole game tree. Lazy-CFR divides the time horizon into segments, i.e., disjoint subsets with consecutive elements, and updates the strategy only at the beginning of each segment and keeps the strategy the same within each segment. By this way we only need to access each infoset at the beginning of each segment.

---

[*]J.Z is the corresponding author.

- We then extend the analysis on the regret of the vanilla CFR in (Burch, 2018) to Lazy-CFR. We show that the regret of Lazy-CFR is almost the same as that of the vanilla CFR. Combining with the analysis on the number of updates in Lazy-CFR, we show that Lazy-CFR converges much faster than the vanilla CFR in theory, with some extra cost of memory.

- Finally, we bound the regret from below. By comparing with the regret lower bound, we show that the regret upper bounds of Lazy-CFR and the vanilla CFR are near-optimal.

We empirically evaluate Lazy-CFR, the vanilla CFR, MC-CFR (Lanctot et al., 2009), CFR+ (Bowling et al., 2017) and MC-CFR+ on the standard benchmarks, Leduc Hold'em (Southey et al., 2005) and heads-up flop hold'em poker (Brown et al., 2019). It is noteworthy that the same idea of Lazy-CFR can also be applied to CFR+, and we name the resulted algorithm Lazy-CFR+. The analysis on Lazy-CFR can be directly applied to Lazy-CFR+. Experiments show that Lazy-CFR and Lazy-CFR+ works well in practice as suggested by the theory.

## 2 NOTATIONS AND PRELIMINARIES

We first introduce the notations and definitions of extensive games and TEGIs. Then we introduce an online learning concept of regret minimization, followed by CFR, which is based on the connection between TEGIs and regret minimization. For clarity, most important notations appeared in this work and their descriptions are listed in the look-up table in appendix A.

### 2.1 EXTENSIVE GAMES

Extensive games (see Osborne & Rubinstein, 1994, pg. 200 for a formal definition) compactly model the decision-making problems with sequential interactions among multiple agents. An extensive game can be represented by a game tree $H$ of histories, where a history is a sequence of actions in the past. Suppose that there are $N$ players participating in an extensive game and let $c$ denote the chance player which is usually used to model the uncertainty in the environment. Let $[N] := \{1, \cdots, N\}$. A player function $P$ is a mapping from $H$ to $[N] \cup \{c\}$ such that $P(h)$ is the player who takes an action after $h$. And each player $i \in [N]$ receives a reward $u^i(h) \in [-1, 1]$ at a terminal history $h$.

Let $\mathcal{A}(h)$ denote the set of valid actions of $P(h)$ after $h$, that is, $\forall a \in \mathcal{A}(h)$, $(h, a) \in H$. Let $A = \max_h |\mathcal{A}(h)|$. A strategy of player $i$ is a function $\sigma^i$ that assigns $h$ a distribution over $\mathcal{A}(h)$ if $P(h) = i$. A strategy profile $\sigma$ consists of the strategy for each player, that is, $\sigma = \{\sigma^1, \cdots, \sigma^N\}$. We will use $\sigma^{-i}$ to refer to all the strategies in $\sigma$ except $\sigma^i$. And we use the pair $(\sigma^i, \sigma^{-i})$ to denote the full strategy profile. In games with imperfect information, actions of other players are partially observable to player $i \in [N]$. So for player $i$, some different histories may not be distinguishable. Thus, the game tree can be partitioned into disjoint information sets (infoset). Let $\mathcal{I}^i$ denote the collection of player $i$'s infosets. We have that two histories $h, h' \in I \in \mathcal{I}^i$ are not distinguishable to player $i$. Thus, $\sigma^i$ must assign the same distribution over actions to all histories in infoset $I \in \mathcal{I}^i$ if $P(I) = i$. With a little abuse of notations, we let $\sigma^i(I)$ denote the strategy of player $i$ on infoset $I \in \mathcal{I}^i$. It is clear that infosets of a player also form a tree, called infoset tree.

We let $\pi_\sigma(h)$ denote the probability of arriving at history $h$. Obviously, we can decompose $\pi_\sigma(h)$ into the product of each player's contribution, that is, $\pi_\sigma(h) = \prod_{[N] \cup \{c\}} \pi_\sigma^i(h)$. Similarly, we define $\pi_\sigma(I) = \sum_{h \in I} \pi_\sigma(h)$ as the probability of arriving at infoset $I$ and let $\pi_\sigma^i(I)$ denote the corresponding contribution of player $i$. Let $\pi_\sigma^{-i}(h)$ and $\pi_\sigma^{-i}(I)$ denote the product of the contributions on arriving at $h$ and $I$, respectively, of all players except player $i$.

With the above notations, a two-player zero-sum extensive game with imperfect information (TEGI) is an extensive game with $N = 2$ and $u^1(h) + u^2(h) = 0$ for all terminal histories.

In game theory, the solution of a game is often referred to a *Nash equilibrium (NE)* (Osborne & Rubinstein, 1994). In this paper, we concern on computing an approximation of an NE, namely an $\epsilon$-**NE** (Nisan et al., 2007). With a little abuse of notations, let $u^i(\sigma)$ denote the expected reward of player $i$ if all players take actions according to $\sigma$. An $\epsilon$-NE is a strategy profile $\sigma$ such that $\forall i \in [N], u^i(\sigma) \geq \max_{\sigma',i} u^i((\sigma'^{,i}, \sigma^{-i})) - \epsilon$. And the $\epsilon$-NE in a TEGI can be efficiently computed by regret minimization; see later in this section.

## 2.2 REGRET MINIMIZATION

Now we introduce *regret*, a core concept in online learning (Cesa-Bianchi & Lugosi, 2006). Many powerful online learning algorithms can be framed as minimizing some kinds of regret, therefore known as regret minimization algorithms. Generally, the regret is defined as follows:

**Definition 1** (Regret). *Consider the case where a player takes actions repeatedly. At each round, the player selects an action $w_t \in \Sigma$, where $\Sigma$ is the set of valid actions. At the same time, the environment selects a reward function $f_t$. Then, the overall reward of the player is $\sum_{t=1}^{T} f_t(w_t)$, and the regret is defined as $R_T = \max_{w' \in \Sigma} \sum_{t=1}^{T} f_t(w') - \sum_{t=1}^{T} f_t(w_t)$.*

One important example of online learning is *online linear optimization* (OLO) in which $f_t$ is a linear function. If $\Sigma$ is the set of distributions over some discrete set, an OLO can be solved by standard regret minimization algorithms, e.g., regret matching (RM) (Blackwell et al., 1956; Abernethy et al., 2011) or AdaHedge (Freund & Schapire, 1997). As CFR employs RM or AdaHedge as a sub-procedure, we summarize them as follows:

**Definition 2** (Online linear optimization (OLO), regret matching (RM) and AdaHedge). *Consider the online learning problem with linear rewards. In each round $t$, an agent plays a mixed strategy $w_t \in \Delta(\mathcal{A})$, where $\Delta(\mathcal{A})$ is the set of distributions, while an adversary selects a vector $c_t \in \mathbb{R}^{|\mathcal{A}|}$. The reward of the agent at this round is $\langle w_t, c_t \rangle$ where $\langle \cdot, \cdot \rangle$ denotes the operator of inner product. The goal of the agent is to minimize the regret: $R_T^{olo} = \max_{w \in \Delta(\mathcal{A})} \sum_{t=1}^{T} \langle w, c_t \rangle - \sum_{t=1}^{T} \langle w_t, c_t \rangle$.*

*Let $R_{T,+}^{olo}(a) = \max(0, \sum_{t=1}^{T} c_t(a) - \sum_{t=1}^{T} \langle w_t, c_t \rangle)$, in RM, $w_{t+1}(a) = R_{t,+}^{olo}(a)/\sum_{a'} R_{t,+}^{olo}(a')$, if $\max_{a'} R_{t,+}^{olo}(a') > 0$, and $w_{t+1}(a) = \frac{1}{|\mathcal{A}|}$ otherwise. According to the result in (Blackwell et al., 1956), RM has the following regret bound:[1]*

$$R_T^{olo} \leq O\left(\sqrt{|\mathcal{A}| \sum_{t=1}^{T} \max_a c_t^2(a)}\right). \tag{1}$$

*Let $s_t(a) = \exp(-\eta_t \sum_{t'=1}^{t} c_{t'}(a))$, AdaHedge picks $w_t(a) = s_t(a)/(\sum_{a'} s_t(a'))$, where $\eta_t$ is the learning rate that can be tuned adaptively (De Rooij et al., 2014). According to (De Rooij et al., 2014), Adahedge has the regret bound $R_T^{olo} \leq O\left(\sqrt{\log |\mathcal{A}| \sum_{t=1}^{T} \max_{a \in \mathcal{A}} c_t^2(a)}\right)$.*

## 2.3 COUNTERFACTUAL REGRET MINIMIZATION (CFR)

CFR is developed on a connection between $\epsilon$-NE and regret minimization. This connection is naturally established by considering repeatedly playing a TEGI as an online learning problem. It is worthy to note that there are two online learning problems in a TEGI, one for each player.

Suppose player $i$ takes $\sigma_t^i$ at time step $t$ and let $\sigma_t = (\sigma_t^1, \sigma_t^2)$. Consider the online learning problem for player $i$ by setting $w_t := \sigma_t^i$ and $f_t^i(\sigma^i) := u^i((\sigma^i, \sigma_t^{-i}))$. The regret for player $i$ is $R_T^i := \max_{\sigma^i} R_T^i(\sigma^i)$, where $R_T^i(\sigma^i) := \sum_{t=1}^{T} u^i((\sigma^i, \sigma_t^{-i})) - \sum_{t=1}^{T} u^i((\sigma_t^i, \sigma_t^{-i}))$. Furthermore, consider the time-averaged strategy $\bar{\sigma}_T^i(I) := \frac{\sum_t \pi_{\sigma_t}^i(I) \sigma_t^i(I)}{\sum_t \pi_{\sigma_t}^i(I)}$. Then, it is well-known that :

**Lemma 1** ((Nisan et al., 2007)). *If $\forall i, \frac{1}{T} R_T^i \leq \epsilon/2$, then $(\bar{\sigma}_T^1, \bar{\sigma}_T^2)$ is an $\epsilon$-NE.*

Specifically, let $\sigma|_{I \to \sigma'(I)}$ denote the strategy generated by modifying $\sigma(I)$ to $\sigma'(I)$ and $u^i(\sigma, I)$ denote the reward of player $i$ conditioned on arriving at the infoset $I$ if the strategy $\sigma$ is executed. (Zinkevich et al., 2008) decomposes the regret $R_T^i$ into the summation of immediate regrets as [2]:

$$R_T^i(\sigma) = \sum_t \sum_{I \in \mathcal{I}^i, P(I)=i} \pi_\sigma^i(I) \pi_{\sigma_t}^{-i}(I)(u^i(\sigma_t|_{I \to \sigma(I)}, I) - u^i(\sigma_t, I)) \tag{2}$$

---

[1] In this work, we use second-order bounds of RM and AdaHedge. These bounds can be easily derived from known results, and we put the derivation of them in Appendix B

[2] (Zinkevich et al., 2008) directly upper bounded $R_T^i$ by the counterfactual regret, i.e., Eq. (3), and omitted the derivation of Eq. (2). So we present the derivation of Eq. (2) in Appendix C.

Further, (Zinkevich et al., 2008) upper bounds Eq. (2) by the **counterfactual regret**:

$$R_T^i(\sigma) \leq \sum_{I \in \mathcal{I}^i, P(I)=i} \left( \sum_t \max(0, \pi_{\sigma_t}^{-i}(I)(u^i(\sigma_t|_{I \to \sigma(I)}, I) - u^i(\sigma_t, I))) \right) \tag{3}$$

For convenience, we call $\pi_{\sigma_t}^{-i}(I)u^i(\sigma_t|_{I \to a}, I)$ the **counterfactual reward** of action $a$ at round $t$. Notice that Eq. (3) essentially decomposes the regret of a TEGI into $O(|\mathcal{I}|)$ OLOs. So that, in each round, we can apply RM directly to each individual OLO to minimize the counterfactual regret. And the original regret $\max_\sigma R_T^i(\sigma)$ is also minimized since the counterfactual regret is an upper bound.

## 3   LAZY-CFR: A LAZY UPDATE ALGORITHM FOR TEGIS

The above CFR procedure that applies RM to solve each OLO has to traverse the whole game tree, which is very time-consuming in large scale games. In this section, we present Lazy-CFR, an efficient CFR algorithm with a lazy update strategy based on the insight that updating the strategy on every infoset is not indispensable. Intuitively, this is because the regret is determined by the norm of the vector of counterfactual reward on each node (see Eq. (1)); and on most nodes, the corresponding norm is very small, since $\pi_{\sigma_t}^{-i}$ is a probability (see Eq. (3)), thereby can be updated in a lazy manner.

### 3.1   LAZY UPDATE FOR OLOS

We start by presenting a lazy update strategy for an OLO in Defn. 2. As illustrated in Fig. 1, a lazy update algorithm for OLOs consists of two steps: (1) It divides the set of time steps $[T]$ into $n$ intervals, that is, $\{t_i, t_i+1, \cdots, t_{i+1}-1\}_{i=1}^n$ where $1 = t_1 < t_2 \cdots < t_{n+1} = T + 1$. (2) It updates $w_t$ at time steps $t = t_i$ for some $i$ and keeps $w_t$ the same within each segment. That is, the OLO with $T$ steps collapses into a new OLO with $n$ steps and accordingly the vector selected by the adversary in the collapsed OLO at time step $j$ is $c'_j = \sum_{t=t_j}^{t_{j+1}-1} c_t$, where $c_t$ is the vector selected by the adversary in the original OLO at time step $t$.

The original OLO

$$c_1 \quad c_2 \quad c_3 \quad c_4 \quad c_5 \quad c_6$$

$$c'_1 = c_1 + c_2 \qquad c'_2 = c_3 + c_4 + c_5 + c_6$$

The collapsed OLO with $t_1 = 1, t_2 = 3, t_3 = 7$

Figure 1: An illustration on RM with lazy update for OLOs. On the top is the standard RM; on the bottom is the RM with lazy update. Their lengths of time are 6 and 2 respectively.

According to the known result in Eq. (1), the regret of the RM with lazy update (Lazy-RM) is bounded by $O(\sqrt{A \sum_{i=1}^n \max_a c_i'^2(a)})$. Thus, if the segmentation rule is under a reasonable design, that is, $\sum_{i=1}^n \max_a c_i'(a)^2 \approx \sum_{j=1}^T \max_a c_j(a)^2$, then the regrets of Lazy-RM and the vanilla RM are similar in amount. As we shall show soon, we can expect a reasonable segmentation rule in CFR.

Though Lazy-RM does not need to update the strategy at each round, a straightforward implementation of Lazy-RM still has a running time of $O(AT)$, which is the same as applying RM directly. This is because we have to compute $\sum_{t=t_i+1}^{t_{i+1}} c_t$. Fortunately, this problem can be addressed in TEGIs by exploiting the structure of the game tree (see Sec. 3.2).

### 3.2   LAZY-CFR

---
**Algorithm 1** Lazy-CFR
---
Input: a constant $\mathcal{B} > 0$.
**while** $t < T$ **do**
    **for all** $i \in \{1, 2\}$ **do**
        $Q = \{I_r\}$ where $I_r$ is the root of the infosets tree.
        **while** $Q$ is not empty. **do**
            Pop $I$ from $Q$.
            Update the strategy on $I$ via RM.
            For $I' \in \gamma(I)$, if $m_t(I') \geq \mathcal{B}$, push $I'$ into $Q$, i.e., $Q = Q \cup \{I'\}$.
        **end while**
    **end for**
    **for all** $h \in H$ such that the strategy on some history after $h$ has been modified. **do**
        Update the reward vector on $h$.
    **end for**
**end while**
Output the time-averaged strategy.

---

We now use lazy update to solve TEGIs. According to Eq. (3), the regret minimization procedure can be divided into $O(|\mathcal{I}|)$ OLOs, one for each infoset. Specifically, for each infoset $I \in \mathcal{I}^i, P(I) = i$, we divide the set of time steps $[T]$ into $n(I)$ segments $\{t_j(I), \cdots, t_{j+1}(I) - 1\}_{j=1}^{n(I)}$, following step

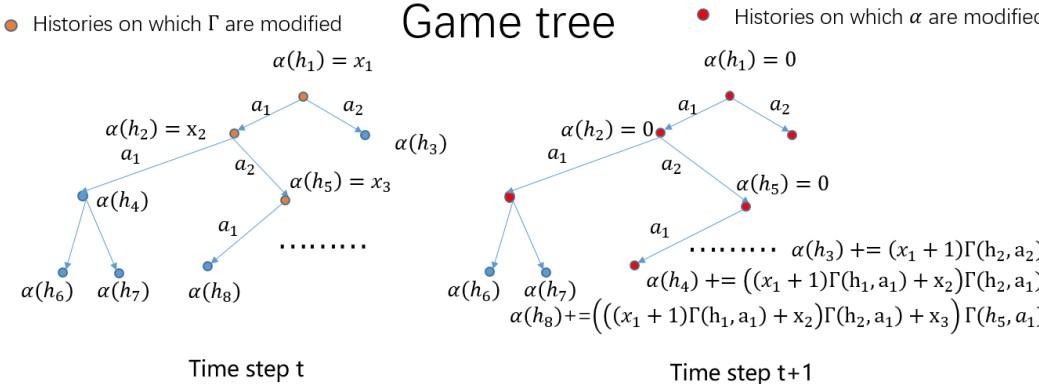

Figure 2: An illustration on how to update $\alpha$. In time step $t$, the strategy on $h_1, h_2, h_5$ is modified with $P(h_1), P(h_2), P(h_5) \neq i$. $\beta$ can be updated in a similar way.

(1). Let $r_j(I, a) = \sum_{t=t_j(I)}^{t_{j+1}(I)-1} \pi_{\sigma_t}^{-i}(I) u^i(\sigma_t|_{I \to a}, I)$ denote the summation of the counterfactual rewards over segment $j$, and let $r_j(I) = [r_j(I, a)]_{a \in \mathcal{A}(I)}$ denote the vector consisting of $r_j(I, a)$. Similar to lazy update for OLOs, we only update the strategy on infoset $I$ at $t_j(I)$. Let $\sigma_j'(I)$ denote the strategy after the $j$-th update on infoset $I$, that is, $\sigma_t(I) := \sigma_j'(I)$ for $t \in [t_j(I), t_{j+1}(I) - 1]$. According to Eq. (1), we can bound the regret of the collapsed OLO on infoset $I$ as $R_T^{lazy}(I) :=$
$$\max_{\sigma \in \Sigma(I)} \sum_{j=1}^{n(I)} \langle \sigma - \sigma_j'(I), r_j(I) \rangle \leq \sqrt{A \sum_{j=1}^{n(I)} \max_a r_j^2(I, a)}.$$

The above procedure is quite straightforward. However, as discussed above, in order to have an efficient implementation, one critical step is to define a proper segmentation rule of each OLO. Below, we present one rule for Lazy-CFR, with which we can: 1) achieve a regret similar in amount to the regret of the vanilla CFR; 2) avoid updating the whole game tree; and 3) compute $r_j(I)$ efficiently.

Specifically, let $\tau_t(I)$ denote the last time step we update the strategy on infoset $I$ before time $t$. We have $\tau_t(h) = \tau_t(I)$ for $h \in I$. Let $m_t(I) := \sum_{\tau=\tau_t(I)+1}^{t} \pi_{\sigma_\tau}^{-i}(I)$ denote the summation of reach probabilities after $\tau_t(I)$, which is contributed by all players except $i$. Let $subt(I)$ denote the subtree rooted at infoset $I$. For convenience, for $I \in \mathcal{I}^i$ if $P(I) = i$, we call $I$ a decision point. Let $\gamma(I)$ denote the set of decision points after $I$ such that every $I'$ in the path from $I$ to $I'' \in \gamma(I)$ is not a decision point. Formally, $\gamma(I)$ is a subset of $subt(I)$ such that $\forall I' \in \gamma(I), P(I') = i$ and $\forall I'' \in subt(I)$, if $I''$ is an ancestor of $I' \in \gamma(I)$, then $P(I'') \neq i$ or $I'' = I$. For convenience, we suppose $P(I_r) = i$ where $I_r$ is the infoset of the empty history. Then, our segmentation rule is defined as a procedure that updates the strategies on infosets recursively as follows: 1) We update the strategy on $I_r$ in every round; 2) For infoset $I$, if we update the strategy on $I$ at time step $t$, the time steps from $\tau_t(I) + 1$ to $t$ forms a segment in the Lazy-RM of $I$. That is, we compute $r_j(I)$ for the corresponding $j$ and then apply RM to $I$; 3) after updating the strategy on infoset $I$, we keep on updating the strategies on the infosets from $\gamma(I)$ with $m_t \geq \mathcal{B}$ where $\mathcal{B} > 0$ is a constant.

Alg. 1 presents an outline of Lazy-CFR. We'll formally analyze its performance in Sec. 4 and now we briefly discuss why Alg. 1 converges faster than CFR. Let's tentatively assume that we can compute $m_t$ and $r_j$ efficiently (See Sec. 3.2.1 for details), then the convergence rate of Lazy-CFR depends on the total number of updates on strategy and its regret. As for the number of updates on strategy, it is obviously upper bounded by $\frac{1}{\mathcal{B}} \sum_{t=1}^{T} \sum_I \pi_{\sigma_t}^{-i}(I)$ which is much smaller than $T|\mathcal{I}|$ as $\pi_{\sigma_t}^{-i}(I)$ is probability. As for the regret, it is easy to see that in Alg. 1, $m_t(I) \leq d(I)\mathcal{B}$ where $d(I)$ is the depth of $I$ in the tree. With $\|r_j(I)\|_2 \leq \max_t m_t(I)$ and Eq. (1), we can upper bound the regret of the Lazy-RM on $I$ by $O(\mathcal{B}d(I)\sqrt{An(I)}) \leq O(\mathcal{B}d(I)\sqrt{AT})$. Therefore, the overall regret, i.e., Eq. (2), of Lazy-CFR is $O(|\mathcal{I}|D\mathcal{B}\sqrt{AT})$. This regret bound is the same to the bound of CFR in (Zinkevich et al., 2008) within a gap of $D\mathcal{B}$ which is usually a logarithm of $|\mathcal{I}|$. In Sec. 4, we further refine this bound and show it is comparable with the best known regret bound of CFR in (Burch, 2018).

### 3.2.1 IMPLEMENTATION

As mentioned in Sec. 3.1, if we compute $m_t(I)$ and $r_j(I)$ directly, we still have to spend $O(AT)$ time on each infoset. In this section, we show how to efficiently implement Alg. 1 by exploiting the structure of the game tree. More specifically, we define data structures (DS) used in the computations of $m_t, r_j$ and then show when and how to update them. Updating these data structures can be implemented by depth first search (DFS) and we only spend $O(1)$ time to update the DSs on each history visited by DFS. The detailed pseudo-code is in Appx E.

W.L.O.G., we consider the DSs used in Lazy-CFR for player $i$. First, we store $u^i(h|\sigma_t)$ in DS $U(h)$ and the strategy profile in $\Gamma(I)$, i.e., $U(h) = u^i(h|\sigma_t), \Gamma^i(I) = \sigma_t^i(I)$ at time step $t$. And how to update $U$ and $\Sigma$ is standard as other CFR algorithms. Recall that our target is to compute $m_t(I) = \sum_{h \in I} m_t(h) = \sum_{h \in I} \sum_{t'=\tau_t(I)+1}^{t} \pi_{\sigma_{t'}}^{-i}(h)$ and $r_j(I, a) = \sum_h r_j(h, a) = \sum_{h \in I} \sum_{t'=\tau_t(I)+1}^{t} \pi_{\sigma_{t'}}^{-i}(h)u^i((h,a)|\sigma_{t'})$ where $(h, a)$ denotes the successor of $h$ after action $a$. It is noteworthy that we do not need to compute $m_t$ and $r_j$ for every history in every round. Instead, we just need to make sure $m_t(h)$ and $r_j(h, a)$ can be computed efficiently when Alg. 1 visits $h$.

Before diving into details, we intuitively explain how to efficiently compute $r_j(h)$ and $m_t(h)$. For $m_t(h)$, suppose we have stored $\sum_{t=\tau_t(h)}^{\tau_t(pa(h))} \pi_t^{-i}(h)$ in some data structure $\alpha(h)$, then we can compute $m_t(h)$ as $m_t(h) = \sum_{h':h \in subt(h')} \alpha(h')\Gamma^{-i}(h', h)$ where $\Gamma(h', h)$ denotes the probability of reaching $h$ from $h'$ contributed by $\Gamma^{-i}$. As for $r_j(h)$, suppose the strategy on histories in $subt(h)$ is never modified during $[\tau_t(h), t]$, then we can compute $r_j(h) = m_t(h)U(h)$ in time $O(1)$, as $U(h)$ is also fixed in $[\tau_t(h), t]$. In Alg. 1, it is possible that the opponent's strategy on some $h'' \in subt(h)$ is modified during $[\tau_t(h), t]$, thus, we need more data structures to maintain the cumulative reach probability and cumulative counterfactual reward.

More specifically, we use two additional DSs $\alpha(h)$ and $\beta(h, a)$. In $\alpha(h)$, we store summations of reach probabilities and we store a summation of counterfactual rewards in $\beta(h, a)$. These DSs should satisfy the following properties:

**Property 1.** *In round $t$, $\alpha, \beta, \hat{\alpha}$ should satisfy:*

1. $\alpha(h) = \sum_{t'=\tau_t(h)+1}^{t_1(h)} \pi_{\sigma_{t'}}^{-i}(h)$, $\beta(h, a) = \sum_{t'=\tau_t(h)+1}^{t_1(h)} \pi_{\sigma_{t'}}^{-i}(h)u^i((h,a)|\sigma_{t'})$ and $\hat{\alpha}(h) = \sum_{t'=t_1(h)+1}^{t_2(h)} \pi_{\sigma_{t'}}^{-i}(h)$ for $\tau_t(h) \leq t_1(h) \leq t_2(h) \leq t$. We will introduce $t_1, t_2$ soon.

2. 1) If $h' = pa(h)$, then $t_1(h') = t_2(h)$; 2) If $h$ is the root, $t_2(h) = t$. So that $[\tau_t(h), t] = \bigcup_{h':h \in subt(h')}[t_1(h') + 1, t_2(h')]$.

3. For all $h' \in subt(h), P(h') \neq i$, the strategy on $h'$ is never modified on time steps between $t_1(h)$ and $t_2(h)$. So that for $t' \in [t_1(h), t_2(h)]$ and $h' \in subt(h)$, $\pi_{\sigma_{t'}}^{-i}(h, h') = \pi_{\Gamma}^{-i}(h, h')$.

Intuitively, $t_1(h)$ is the last time step that $U(h)$ is modified and $t_2(h)$ is the last time step that the strategy on $pa(h)$ is modified. Before discussing how to update these DSs, we discuss how to use them to compute $m_t$ and $r_j$. It is easy to check that Eq. (4) is true with Prop. 1, we have:

$$m_t(h) = \sum_{h':h \in subt(h')} \alpha(h')\pi_{\Gamma}^{-i}(h', h), \quad r_j(h, a) = \beta(h, a) + (m_t(h) - \alpha(h))U((h, a)) \quad (4)$$

where $\pi_{\Gamma}^{-i}(h', h)$ denotes the probability from $h'$ to $h$ contributed by $-i$ with strategy profile $\Gamma$. We postpone the derivation of Eq. (4) to Appx. E. With the fact that $m_t(h) = m_t(pa(h))\pi_{\Gamma}^{-i}(pa(h), h) + +\alpha(h)$ where $pa(h)$ is the parent of $h$, computing $m_t$ only spends $O(1)$ time on each node as when visiting $h$, $m_t(pa(h))$ must have been computed in DFS.

The last challenge is how to update these DSs to satisfy the desired properties in Prop 1. Specifically, let $S_{1,t}$ denote the set of histories such that if $h \in S$, the strategy on $subt(h)$ is modified at round $t$ and $S_{2,t} = \{h : h \notin S_{1,t}, pa(h) \in S_{1,t}\}$. As illustrated in Fig. 2, we only update DSs on $S_t = S_{1,t} \cup S_{2,t}$ as:

**Update on the DSs:** 1) If $\Gamma$ on the infoset of $h$ is modified, set $\alpha(h) = \beta(h, a) = 0$; else 2) for $h \in S_t$, and set $\alpha(h) = m_t(h), \beta(h, a) = m_t(h)U((h, a))$.

With above update rule, it is easy to check that if Prop 1 is true at $t$, then it is still true at $t+1$.

***Lazy-CFR+** and **Lazy-LCFR***: We can also apply lazy update to CFR+ (Bowling et al., 2017) and LCFR (Brown & Sandholm, 2019a), which are improvements of CFR. To get Lazy-CFR+ and Lazy-LCFR, we only need to replace RM by the corresponding OLO solvers, and use their methods of computing time-averaged strategy as in (Bowling et al., 2017) and (Brown & Sandholm, 2019a) respectively.

## 4 THEORETICAL ANALYSIS

We now present the theoretical results, starting with the regret for members of CFR with lazy update.

### 4.1 REGRET UPPER BOUND

We extend the regret analysis on the vanilla CFR in (Burch, 2018) to the members of CFR with lazy update. Specifically, let $\xi^i(\sigma) = \sum_{d=1}^{D} \sqrt{\sum_{I:d(I)=d} \pi_\sigma^i(I)}$, $\xi = \max_{i,\sigma} \xi^i(\sigma)$ and $\eta(\sigma) = \xi^i(\sigma)\sqrt{A \max_{I,j}(\sum_{t=t_j(I)+1}^{t_{j+1}(I)} \pi_{\sigma_t}^{-i}(I))}$ which are parameters depending on the structure of the game and segmentation rule. Thm. 1 provides a regret bound for a CFR algorithm with an arbitrary segmentation rule. By applying Thm. 1, we can get the regret bound of CFR and Lazy-CFR which are comparable with Corollary 2 in (Burch, 2018).

**Theorem 1.** *The regret of CFR with lazy update can be bounded as $R_T^i(\sigma) \leq O(\sqrt{T}\eta(\sigma))$.*

**Lemma 2.** *With RM, the regret of the vanilla CFR is bounded by $O(\xi\sqrt{AT})$ and the regret of Lazy-CFR is bounded by $O(\xi\sqrt{DAT})$.*

*Proof.* It is easy to see that for the vanilla CFR, we have $\eta(\sigma) \leq \xi^i(\sigma)\sqrt{A}$ and for Lazy-CFR, we have $\eta(\sigma) \leq \xi^i(\sigma)\sqrt{AD}$. With Thm. 1, we get the regret bounds. □

### 4.2 TIME AND SPACE COMPLEXITY

With the implementation in Sec. 3.2.1 and Appx E, the running time of Lazy-CFR is bounded by $O(\sum_t |S_t|)$. Obviously, $\sum_t |S_t| = O(\max_\sigma \sum_t)\pi_\sigma^{-i}(h)$. Thus, we can bound Lazy-CFR's time complexity as:

**Lemma 3** (Time complexity). *The time complexity of Alg. 1 is $O(T \max_\sigma \sum_h \pi_\sigma^{-i}(h))$.*

To show how small $\max_\sigma \sum_I \pi_\sigma^{-i}(I)$ is, we make the following mild assumption which leads to Corollary 1.

**Assumption 1.** *1, If $P(h) = i$, then $P((h,a)) \neq i$; 2, The tree of infosets for each player is a full $A$-ary tree; 3, Every infoset in the tree of infosets is corresponding to $n$ nodes in the history tree.*

**Corollary 1.** *If a TEGI satisfies Assumption 1, then $\forall \sigma, \sum_{h \in \mathcal{I}^i} \pi_\sigma^{-i}(h) = O(n\sqrt{|\mathcal{I}^i|})$.*

According to Lemmas 2 3 and 1, the regret of Lazy-CFR is about $O(\sqrt{D})$ times larger than that of CFR, whilst the running time is about $O(\sqrt{|\mathcal{I}|})$ times faster than CFR per round under Assumption 1. Thus, according to Lemmas 1, 2 and with a bit algebra calculation, we know that Alg. 1 is about $O(\sqrt{|\mathcal{I}|}/D)$ times faster than the vanilla CFR to achieve the same approximation error. The improvement is significant in large scale TEGIs.

**Space complexity**: A potential limitation of Lazy-CFR is that its space complexity is $O(|H|)$ as we have to store the data structures $\alpha, \beta, \hat{\alpha}$ on histories. In contrast, the space complexity is $O(|\mathcal{I}|)$ for some popular implementation of CFR. In heads-up flop hold'em poker (FHP) (Brown et al., 2019), we have $|H| \approx 10^{12}$ and $|\mathcal{I}| \approx 10^9$. Accordingly, Lazy-CFR needs about 10 TB to store the data structures while CFR needs about 10 GB, which is still affordable in common storage systems. Note that we do not optimize the space complexity in this work. It is worth of a systematical investigation to derive an algorithm as fast as Lazy-CFR without additional memories, e.g., by designing a better segmentation rule as well as a better implementation.

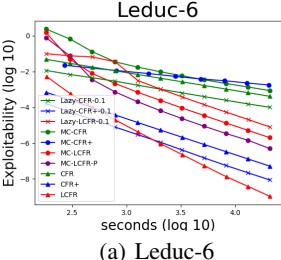
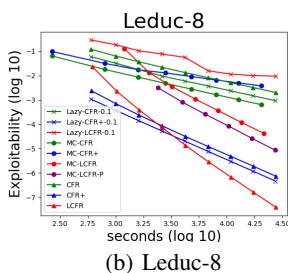
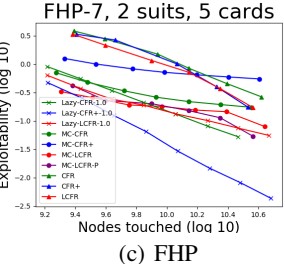

(a) Leduc-6        (b) Leduc-8        (c) FHP

Figure 3: Convergence for Lazy-(CFR, CFR+, LCFR), MC-(CFR, CFR+, LCFR), CFR, CFR+, LCFR and MC-LCFR-P.

### 4.3 REGRET LOWER BOUND

In the analysis of lower bound, we consider the standard adversarial setting in online learning, in which an adversary selects $\sigma_t^{-i}$ and a reward function $u_t^i : Z \to [-1, 1]$ where $Z$ is the set of terminal nodes in the infoset tree of player $i$. Thus, to get a lower bound, we need to explicitly construct $\sigma_t^{-i}$ and $u_t^i(I), I \in Z$. This setting is consistent with the analysis of the regret upper bound, in which we do not make any assumption on how $u_t^i, \sigma_t^{-i}$ changes over rounds. Intuitively, by extending the lower bound analysis of OLO (Cesa-Bianchi & Lugosi, 2006) to infoset tree structured problem, we bound Eq. 2 from below as in Thm. 2 and for the proof, see Appendix D.

**Theorem 2.** *For an algorithm $\mathbb{A}$ generating $\sigma_t^i$ given the history in the past, let $R_{T,\mathbb{A}}^i$ denote the regret of $\mathbb{A}$ in the first $T$ rounds, we have* $\lim_{A \to \infty} \lim_{T \to \infty} \min_{\mathbb{A}} \max_{\pi_{\sigma_t}^{-i}, u_t} \frac{R_{T,\mathbb{A}}^i}{\xi^i \sqrt{T/D \log A}} \geq 1.$

By comparing the regret lower bound in Theorem 2 and the regret upper bounds of CFR and Lazy-CFR as in Lem 2, we can see that the regret of CFR and Lazy-CFR are both near-optimal.

## 5 RELATED WORK

**Monte-Carlo and pruning-based methods:** There are several variants of CFR which attempt to avoid traversing the whole game tree at each round. Monte-Carlo based CFR (MC-CFR) (Lanctot et al., 2009; Burch N, 2012) uses Monte-Carlo sampling to avoid updating the strategy on infosets with small probability of arriving at. Pruning-based variants (Brown & Sandholm, 2017a; 2015) skip the branches of the game tree if they do not affect the regret, but their performance can deteriorate to the vanilla CFR in the worst case. And dynamic thresholding (Brown et al., 2017) directly prunes the branches with small reach probabilities. In this work, we do not compare with pruning-based method (Brown & Sandholm, 2019b) since the pruning technique is orthogonal to lazy update.

**Existing analyses of regret**: Lanctot et al. (2009); Burch N (2012); Burch (2018) refined the regret upper bound of CFR. Our analysis is essentially an extension of the regret analysis on the vanilla CFR in (Burch, 2018) to other variants of CFR with lazy-update.

## 6 EXPERIMENT

In this section, we empirically evaluate our algorithm against existing CFR variants. We measure the exploitability of these algorithms. The exploitability of a strategy $(\sigma^1, \sigma^2)$ can be interpreted as the approximation error to the Nash equilibrium. The exploitability is defined as $\max_{\sigma'^1} u^1((\sigma'^1, \sigma^2)) + \max_{\sigma'^2} u^2((\sigma^1, \sigma'^2))$.

Experiments are conducted on variants of two common benchmarks in imperfect-information game solving: (1) the Leduc hold'em (Southey et al., 2005) which is a simplfied version of the heads-up no-limit hold'em poker with 6 cards; and (2) heads-up flop hold'em poker (Brown et al., 2019) (FHP) which is similar to heads-up no-limit Texas hold'em poker without the last two rounds of betting.

In our experiments of Leduc Hold'em poker, we restrict players not to bid more than 6 or 8 times the big blind. The number of infosets in the generated game trees are about 40000 and 455000 for each player, respectively. We run each algorithm on each Leduc Hold'em game for 30000 seconds. In the experiments of FHP, to control the size of the game, we consider a simplified game such that in the

deck, there are 2 suits and 5 cards in each suit. And we restrict players not to bid more than 7 times the big blind. The size of the game is about $10^9$. We measure algorithms by the number of nodes touched which is independent with implementation and hard-ware.

We empirically compare Lazy-(CFR,CFR+,LCFR) with existing methods, including the vanilla CFR, CFR+, LCFR, MC-(CFR,CFR+,LCFR) and MC-LCFR with negative regret-pruning (MC-LCFR-P) which was used in developing Pluribus (Brown & Sandholm, 2019b). In our experiments, we use RM (RM+) as the OLO solver. In Lazy-(CFR,CFR+,LCFR), we set $\mathcal{B} \in \{0.1, 1.0\}$. We evaluate the CFR, CFR+, LCFR variants which prunes the histories with $\pi_{\sigma_t}^{-i}(h) = 0$ in the recursive tree walk as they don't affect the regret. For MC-(CFR,CFR+,LCFR), we use the external-sampling scheme. In the experiments on MC-LCFR-P, we use the following parameters: we run MC-LCFR in the first 20 minutes; after that, in each iteration, we run MC-LCFR with probability 0.05 and run MC-LCFR-P with probability 0.95; in MC-LCFR-P, we prune those branches with average regret less than $-2$ times the big blind.

Fig. 3 presents the results. We can see that the performance of Lazy-CFR(+) has a similar performance to CFR(+) on the variants of Leduc Hold'em. This is because in the experiments of CFR(+) on Leduc Hold'em, there are a large portion of histories with $\pi_t^{-i}(h) = 0$ on average. And on the larger game, Lazy-CFR(+) significantly outperforms CFR(+). The performance of Lazy-LCFR is much worse than LCFR on all games. This might be because our segmentation rule is designed for OLO with a uniform weight in each iteration. And LCFR assigns more weights on later iterations.

## 7    CONCLUSIONS

In this work, we propose a new framework to develop efficient variants of CFR with an analysis shows that our algorithm is provably faster than the vanilla CFR. The final algorithm runs fast in practice, but with some extra cost on space complexity. It is worth of a systematical study to reduce the space complexity.

## ACKNOWLEDGEMENT

This work was supported by the National Key Research and Development Program of China (No. 2017YFA0700904), NSFC Projects (Nos. 61620106010, U19B2034, U1811461), Beijing NSF Project (No. L172037), Beijing Academy of Artificial Intelligence (BAAI), Tsinghua-Huawei Joint Research Program, a grant from Tsinghua Institute for Guo Qiang, Tiangong Institute for Intelligent Computing, the JP Morgan Faculty Research Program and the NVIDIA NVAIL Program with GPU/DGX Acceleration.

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

# A   THE TABLE OF NOTATIONS

We provide a table of some important notations in our paper in this section.

| Notation | Explanation |
|---|---|
| $[n]$ | the set $\{1, 2, \cdots, n\}$ |
| $[a, b]$ | the set $\{a, a+1, \cdots, b\}$ |
| $H$ | The tree of histories |
| $I$ | Infomation set |
| $D$ | the depth of the game tree |
| $d(I)$ and $d(h)$ | the depths of $I$ and $h$ |
| $c$ | the chance player |
| $h$ | history |
| $Z \subseteq H$ | the set of leaves of the game tree |
| $(h, a)$ | the successor of $h$ after action $a$ |
| $(I, a)$ | the successor of $I$ after action $a$ |
| $\mathcal{I}^i, \mathcal{I}$ | $\mathcal{I}^i$ is the infoset tree of player $i$ and $\mathcal{I} = \mathcal{I}^1 \cup \mathcal{I}^2$ |
| $P$ | The player function, i.e., $P(h)$ and $P(I)$ are the players who take actions on $h$ and $I$, respectively. |
| $\sigma^i$ | the strategy of player $i$; |
| $\sigma$ | the strategy of all players; |
| $\sigma^{-i}$ | the strategy of all players except player $i$ |
| $\sigma(h), \sigma(I)$ | $\sigma(h) = \sigma^{P(h)}(h)$ and $\sigma(I) = \sigma^{P(I)}(I)$ |
| $\pi_\sigma(h), \pi_\sigma(I)$ | the reach probability of $h$ and $I$ contributed by all players |
| $\pi_\sigma^i(h), \pi_\sigma^i(I)$ | the reach probability of $h$ and $I$ contributed by player $i$ |
| $\pi_\sigma^{-i}(h), \pi_\sigma^{-i}(I)$ | the reach probability of $h$ and $I$ contributed by all players except $i$ |
| $\pi^{-i}(h', h)$ | the reach probability from $h'$ to $h$ contributed by all players except $i$ if $h \in subt(h')$ |
| $u^i(h), h \in Z$ | the reward received by player $i$ at leaf $h$ |
| $u^i(\sigma)$ | the reward received by player $i$ with strategy profile $\sigma$ |
| $u^i(h|\sigma)$ | the reward received by player $i$ at $h$ with strategy profile $\sigma$ |
| $\sigma|_{I \to \sigma'(I)}$ | the strategy profile generated from $\sigma$ by replacing $\sigma(I)$ by $\sigma'(I)$ |
| $pa(h), pa(I)$ | the parents of $h$ and $I$ respectively |
| $subt(h), subt(I)$ | the subtree rooted at $h$ and $I$ respectively |
| $\gamma(I), I \in \mathcal{I}^i$ | a subset of $subt(I)$ such that for all $I' \in \gamma(I)$ then $P(I') = i$; consider $I'' \in subt(I)$, if $I''$ is an ancestor of $I'$, then $P(I'') \neq i$ |
| $\langle a, b \rangle$ | the inner product of vectors $a$ and $b$ |
| $w_t$ | the action selected by the player at time step $t$ |
| $c_t$ | the reward vector at time step $t$ |
| $R_T^{olo}$ | the regret of the OLO, i.e., $R_T^{olo} = \max_{w \in \Delta(\mathcal{A})} \sum_{t=1}^T \langle w, c_t \rangle - \sum_{t=1}^T \langle w_t, c_t \rangle$ |
| $\Delta(\mathcal{A})$ | the set of distributions over set $\mathcal{A}$ |
| $R_T^i(\sigma^i)$ | $R_T^i(\sigma^i) = \sum_{t=1}^T u^i(\sigma^i, \sigma_t^{-i}) - \sum_{t=1}^T u^i((\sigma_t^i, \sigma_t^{-i}))$ |
| $R_T^i$ | the regret of player $i$, i.e., $R_T^i = \max_{\sigma^i} R_T^i(\sigma^i)$ |
| $\bar{\sigma}_T^i$ | the time-averaged strategy with $\bar{\sigma}_T^i(I) = \frac{\sum_{t=1}^T \pi_{\sigma_t}^i(I)\sigma_t(I)}{\sum_{t=1}^T \pi_{\sigma_t}^i(I)}$ |
| $\tau_t(I), \tau_t(h)$ | the last time step we update the strategy on infoset $I$ (history $h$) before time $t$ |
| $m_t(h)$ | the sum of reach probabilities, $m_t(h) = \sum_{t'=\tau_t(h)+1}^t \pi_{\sigma_{t'}}^{-i}(h)$ |
| $m_t(I)$ | $m_t(I) = \sum_{h \in I} m_t(h)$ |
| $r_j(h, a)$ | the sum of counterfactual rewards over the $j$-th segment of $h$ |
| $r_j(I, a)$ | $r_j(I, a) = \sum_{h \in I} r_j(h, a)$ |
| $U(h)$ | data structure with $U(h) = u^i(h|\sigma_t)$ at time step $t$ |
| $\Sigma$ | data structure with $\Sigma(I) = \sigma_t(I)$ at time step $t$ |
| $\alpha, \beta, \hat{\alpha}$ | data structures such that, at time step $t$, $m_t(h) = \alpha(h) + \sum_{h':h \in subt(h)} \hat{\alpha}(h)\pi_\Gamma^{-i}(h)$ and $r_j(h, a) = \beta_j(h, a) + U((h, a)) \sum_{h':h \in subt(h')} \hat{\alpha}(h)\pi_\Gamma^{-i}(h)$ |
| $\xi^i(\sigma)$ | a parameter depends on the structure of the game tree with $\xi^i(\sigma) = \sum_{I \in \mathcal{I}^i, P(I)=i} \pi_\sigma^i(I)$ |
| $\xi$ | $\xi = \max_{i,\sigma} \xi^i(\sigma)$ |
| $\eta(\sigma)$ | a parameter depends on the segmentation rule with $\eta(\sigma) = A\xi^i(\sigma) \max_{I,j}(\sum_{t=t_j(I)+1}^{t_{j+1}(I)} \pi_{\sigma_t}^{-i}(I))$ |

# B    DERIVATION OF REGRET BOUNDS OF RM AND ADAHEDGE

The regret upper bound of RM and AdaHedge used in this work can be easily derived from previous works. We include their derivations only for completeness.

*Derivation of Eq.* (1). The standard analysis of RM in Cor 2.1 on pg. 13 in (Cesa-Bianchi & Lugosi, 2006) shows the regret of RM is bounded by:

$$R_T^{olo} \le O\left(\sqrt{\sum_{t=1}^{T}\sum_{a=1}^{|\mathcal{A}|}\left(\sum_{a'=1}^{|\mathcal{A}|} w_t(a')c_t(a') - c_t(a)\right)^2}\right)$$

Obviously, we have $|\sum_{a'=1}^{|\mathcal{A}|} w_t(a')c_t(a')| \le \max_a |c_t(a)|$ since $w_t$ is a vector of probability. We have:

$$\sum_{a=1}^{|\mathcal{A}|}\left(\sum_{a'=1}^{|\mathcal{A}|} w_t(a')c_t(a') - c_t(a)\right)^2 \le \sum_a 4\max_a c_t^2(a) = 4|\mathcal{A}|\max_a c_t^2(a).$$

We finished the proof.                                                                     □

*Derivation of the regret bound of AdaHedge.* Let $v_t$ denote the variance of reward if we take a random action according to the distribution $w_t$ defined in AdaHedge. Theorem 6 in (De Rooij et al., 2014) provides a second-order bound as:

$$R_T^{olo} \le \kappa\sqrt{\log|\mathcal{A}|\sum_t v_t} + \frac{4}{3}\log|\mathcal{A}| + 2$$

where $\kappa$ is a constant. It is known that for a random variable $x \in [a, b]$, its variance is no more than $(a - b)^2/4$. Thus, $v_t \le \max_a c_t^2(a)$. We finished the proof.                    □

## C  PROOF OF THEOREM 1

We now extend the proof of the Theorem 1 in (Burch, 2018) to CFR with lazy update. We first introduce some additional terminology. Consider $B \subseteq \mathcal{I}^i$, let $\xi(B) = \max_{\sigma^{-i}} \sum_{I \in B} \sum_{h \in I} \pi_{\sigma^{-i}}^{-i}(h)$.

Now we prove Theorem 1.

*Proof.* Let $\kappa$ denote the constant involved in Eq. (1). With Eq. (2) and the regret bound of RM, we have

$$\frac{1}{T} R_T^i(\sigma) = \frac{1}{T} \sum_t \sum_{I \in \mathcal{I}^i, P(I) = i} \pi_\sigma^i(I) \pi_{\sigma_t}^{-i}(I) (u^i(\sigma_t|_{I \to \sigma(I)}, I) - u^i(\sigma_t, I))$$

$$\leq \kappa \sum_{I \in \mathcal{I}^i, P(I) = i} \left( \pi_\sigma^i(I) \sqrt{\sum_{j=1}^{n(I)} A \max_a (r_j(I, a))^2 / T^2} \right)$$

And then apply Jensen's inequality and with some calculations, we have

$$\frac{1}{T} R_T^i(\sigma)$$

$$\leq \kappa \sum_{I \in \mathcal{I}^i} \left( \pi_\sigma^i(I) \sqrt{A \sum_{j=1}^{n(I)} \max_a r_j^2(I, a) / T^2} \right)$$

$$= \kappa \sum_{d=1}^{D} \left( \frac{\sum_{I', t, d(I) = d} \pi_\sigma^i(I') \pi_{\sigma_t}^{-i}(I')}{\sum_{I', t, d(I') = d} \pi_\sigma^i(I') \pi_{\sigma_t}^{-i}(I')} \right) \cdot \sum_{I : d(I) = d} \left( \pi_\sigma^i(I) \sqrt{A \sum_{j=1}^{n(I)} \max_a r_j^2(I, a) / T^2} \right)$$

$$= \kappa \sum_{d=1}^{D} \left( \sum_{I', t, d(I') = d} \pi_\sigma^i(I') \pi_{\sigma_t}^{-i}(I') \right) \sum_{I : d(I) = d} \left( \frac{\pi_\sigma^i(I)}{\sum_{I', t, d(I') = d} \pi_\sigma^i(I') \pi_{\sigma_t}^{-i}(I')} \right) \left( \sqrt{A \sum_{j=1}^{n(I)} \max_a r_j^2(I, a) / T^2} \right)$$

$$= \kappa \sum_{d=1}^{D} \left( \sum_{I', t, d(I') = d} \pi_\sigma^i(I') \pi_{\sigma_t}^{-i}(I') \right) \sum_{I : d(I) = d} \left( \frac{\pi_\sigma^i(I) \sum_t \pi_{\sigma_t}^{-i}(I)}{\sum_{I', t, d(I') = d} \pi_\sigma^i(I') \pi_{\sigma_t}^{-i}(I')} \right) \left( \sqrt{A \frac{\sum_{j=1}^{n(I)} \max_a r_j^2(I, a)}{\left( \sum_t \pi_{\sigma_t}^{-i}(I) \right)^2 T^2}} \right)$$

$$= \kappa \sum_{d=1}^{D} \left( \sum_{I', t, d(I') = d} \pi_\sigma^i(I') \pi_{\sigma_t}^{-i}(I') \right) \sum_{I : d(I) = d} \left( \frac{\sum_t \pi_\sigma^i(I) \pi_{\sigma_t}^{-i}(I)}{\sum_{I', t, d(I') = d} \pi_\sigma^i(I') \pi_{\sigma_t}^{-i}(I')} \right) \left( \sqrt{A \frac{\sum_{j=1}^{n(I)} \max_a r_j^2(I, a)}{\left( \sum_t \pi_{\sigma_t}^{-i}(I) \right)^2 T^2}} \right)$$

$$\leq \kappa \sum_{d=1}^{D} \left( \sum_{I', t, d(I') = d} \pi_\sigma^i(I') \pi_{\sigma_t}^{-i}(I') \right) \left( \sqrt{A \sum_{I : d(I) = d} \left( \frac{\sum_t \pi_\sigma^i(I) \pi_{\sigma_t}^{-i}(I)}{\sum_{I', t, d(I') = d} \pi_\sigma^i(I') \pi_{\sigma_t}^{-i}(I')} \right) \cdot \left( \frac{\sum_{j=1}^{n(I)} \max_a r_j^2(I, a)}{\left( \sum_t \pi_{\sigma_t}^{-i}(I) \right)^2 T^2} \right)} \right)$$

The Jensen's inequality applies here, in which we move the term of probability outside the square root into it.

$$= \kappa \sum_{d=1}^{D} \left( \sqrt{A \left( \sum_{I', t, d(I') = d} \pi_\sigma^i(I') \pi_{\sigma_t}^{-i}(I') \right) \sum_{I : d(I) = d} \left( \pi_\sigma^i(I) \frac{\sum_{j=1}^{n(I)} \max_a r_j^2(I, a)}{\sum_t \pi_{\sigma_t}^{-i}(I) T^2} \right)} \right)$$

$$\leq \kappa \sum_{d=1}^{D} \left( \sqrt{A \frac{1}{T} \sum_{I : d(I) = d} \pi_\sigma^i(I) \frac{\sum_{j=1}^{n(I)} \max_a r_j^2(I, a)}{\sum_{t=1}^{T} \pi_{\sigma_t}^{-i}(I)}} \right)$$

The last inequality utilizes the fact that $\sum_{I', t, d(I') = d} \pi_\sigma^i(I') \pi_{\sigma_t}^{-i}(I') \leq T$ as $\pi_{\sigma_t}^{-i}(I) \pi_\sigma^i(I)$ is the probability of arriving at infoset $I$ under strategy $(\sigma^i, \sigma_t^{-i})$ and there is at most 1 infoset been arrived at each level.

It is easy to see that $\max_a r_j^2(I,a) \le (\sum_{t=t_j(I)+1}^{t_{j+1}(I)} \pi_{\sigma_t}^{-i}(I))^2$. With straight-forward computations, we finish the proof:

$$A \sum_{I:d(I)=d} \pi_{\sigma^i}^i(I) \frac{\sum_{j=1}^{n(I)} (\sum_{t=t_j(I)+1}^{t_{j+1}(I)} \pi_{\sigma_t}^{-i}(I))^2}{\sum_t \pi_{\sigma_t}^{-i}(I)}$$

$$\le A \sum_{I:d(I)=d} \pi_{\sigma^i}^i(I) \max_j (\sum_{t=t_j(I)+1}^{t_{j+1}} \pi_{\sigma_t}^{-i}(I)), \text{ this is because } \frac{\sum_i x_i^2}{\sum_i x_i} \le \max_i x_i \text{ if } x_i \ge 0, \sum_i x_i > 0$$

$$\le A \left( \sum_{I \in \mathcal{I}^i} \pi_\sigma^i(I) \right) \max_{I,j} \left( \sum_{t=t_j(I)+1}^{t_{j+1}} \pi_{\sigma_t}^{-i}(I) \right)$$

$\square$

## D    THE REGRET LOWER BOUND ANALYSIS

Recall that the analysis on the regret upper bound is under the standard adversarial setting in online learning, that is, it does not depend on how the opponent's strategy and the utility vary over time steps. So we here make the same adversarial assumption that there is an adversary choose both $\sigma_t^{-i}$ and $u_t^i$. It is worthy to note that in our construction of the adversary, the utility function may also vary over time.

For convenience, let $-i$ denote all the players except $i$, and let $\zeta = \{\hat{\sigma}^i : \hat{\sigma}^i = \arg\max_{\sigma^i} \sum_{d'=1}^D \sum_{I \in \mathcal{I}^i, P(I)=i} \pi_{\sigma^i}^i(I)\}$. Let $\mathcal{D} := \{I \in \mathcal{I}^i : \exists \sigma^i \in \zeta, \pi_{\sigma^i}^i(I) > 0\}$. It can be shown that $\mathcal{D}$ forms a subtree of $\mathcal{I}^i$. Intuitively, our construction on $\sigma_t^{-i}$ and $u_t^i$ can be divided into two stages:

1. For $I \in Z, I \notin \mathcal{D}, u_t^i(I) = -1$ for all $t$. This enforces player $i$ take actions on $\mathcal{D}$, otherwise, it will always receive reward $-1$.

2. In each round $t$, for $I \in \mathcal{D}, I \notin Z, P(I) \ne i$, we first generate a random variable $a(I) \sim$ Multinomial$(1, \frac{1}{\mathcal{A}(I)})$, [3] and then set $\sigma_t^{-i}(I, a(I)) = 1$, and $\sigma_t^{-i}(I, a) = 0$ for $a \ne a(I)$. Intuitively, this step separates $R_T^i$ into $O((\xi^i)^2)$ isolated OLOs, each of which is of $A$ actions and would be repeated for about $O(T/(\xi^i)^2)$ rounds, since only one of them will be triggered on in each time step according to our construction on $\sigma_t^{-i}$. Thus, combined with the lower bound proved by (Cesa-Bianchi & Lugosi, 2006), each OLO incurs a regret of about $\Omega(\sqrt{T \log A}/\xi^i)$, and we can informally provide a lower bound of $\Omega((\xi^i)^2/\xi^i \sqrt{T \log A}) = \Omega(\xi^i \sqrt{T \log A})$, which is formally described in Theorem 2.

Before proving Theorem 2, we first address some trivial cases and show some intuitions on the way we construct the worst case of $\sigma_t^{-i}$ and $u_t^i$.

Let $\varphi^i(\sigma) = \sum_{I \in \mathcal{I}^i} \pi_\sigma^i(I)$. It is easy to see that $\varphi^i(\sigma) > (\xi^i(\sigma))^2/D$. And we are going to show that $\lim_{A\to\infty} \lim_{T\to\infty} \min_{\mathbb{A}} \max_{\pi^{-i},u} R_{T,\mathbb{A}}^i / \sqrt{\varphi^i T \log A} \ge 1$. To start, we make an implicit assumption that $|\mathcal{A}(I)| \ge 2, \forall I \in \mathcal{I}^i, P(I) = i$. Otherwise we can merge these infosets as we have no choice but choose the only action, which contributes nothing to the regret. In addition, we assume $i$ and $-i$ take actions alternatively.

We now show that it is sufficient to focus on the subtree $\mathcal{D}$ (Note that $\mathcal{D}$ is a subtree rooted at $I_r$). By setting $u_t(I) = -1$ for $I \in Z$ and $I \notin \mathcal{D}$, we know that player $i$ will not take actions to go out of $\mathcal{D}$. And if player $-i$ goes out of $\mathcal{D}$, this round will contribute nothing to the regret as player $i$ will always receive reward $-1$.

Moreover, we assume $P(pa(I)) = i, \forall I \in Z$. Otherwise, we can merge the subtree rooted at $pa(I), I \in Z$ into one single leaf node (i.e. an infoset $I \in Z$ after the mergence), and $P(pa(I)) = i, \forall I \in Z$ in the new merged infoset tree.

---

[3] This step is informal here. See later in this section for a formal construction.

The following property will be useful in our proof.

**Lemma 4.** *Let $\varphi_{\mathcal{D}}^i(\sigma^i) = \sum_{I \in \mathcal{D}} \pi_{\sigma^i}^i(I)$. Moreover, let $\zeta' = \{\sigma^i : \sigma^i(I, a) = 0, \text{ for } P(I) = i, I \in \mathcal{D}, (I, a) \notin \mathcal{D}\}$. Then $\forall \sigma_{\mathcal{D}}^i, (\sigma_{\mathcal{D}}^i)' \in \zeta'$, we have that $\varphi_{\mathcal{D}, \sigma_{\mathcal{D}}^i}^i = \varphi_{\mathcal{D}, (\sigma_{\mathcal{D}}^i)'}^i$.*

*Proof.* To start, let $\varphi_{\mathcal{D}}^i(\sigma_i, I) = \frac{1}{\pi_{\sigma_i}^i(I)} \sum_{I' \in subt(I), P(I') = i} \pi_{\sigma^i}^i(I'), I \in \mathcal{D}$, where $subt(I)$ means the subtree rooted at $I$ (Note that it's possible that $I' \in subt(I), I' \notin \mathcal{D}$). Let $\varphi_{\mathcal{D}}^i(I) = \max_{\sigma^i} \varphi_{\mathcal{D}}^i(\sigma^i, I)$, and $\sigma^{i,*} = \arg\max_{\sigma^i} \varphi_{\mathcal{D}}^i(\sigma^i, I)$. We first show that, $\forall I \in \mathcal{D}, P(I) = i$, $\varphi_{\mathcal{D}}^i(J^i(I, a_j)) = \varphi_{\mathcal{D}}^i(J^i(I, a_k)), j, k \in |\mathcal{A}_{\mathcal{D}}(I)|$ where $\mathcal{A}_{\mathcal{D}}(I)$ consists of $a \in \mathcal{A}(I)$ such that $J^i(I, a) \in \mathcal{D}$.

With the definition of $\mathcal{D}$, $\forall I \in \mathcal{D}, \exists \sigma_0^i \in \zeta$, s.t. $\pi_{\sigma_0^i}^i(I) > 0$. If $\varphi_{\mathcal{D}}^i(J^i(I, a_j)) \neq \varphi_{\mathcal{D}}^i(J^i(I, a_k))$, W.L.O.G. we assume $\varphi_{\mathcal{D}}^i(J^i(I, a_j)) < \varphi_{\mathcal{D}}^i(J^i(I, a_k))$, then $\forall \sigma_0^i$ such that $\pi_{\sigma_0^i}^i(I) > 0$, we have

$$\varphi_{\mathcal{D}}^i(\sigma_0^i, I) = 1 + \sum_{a \in \mathcal{A}(I)} \sigma_0^i(I, a) \varphi_{\mathcal{D}}^i(\sigma_0^i, J^i(I, a))$$

$$\leq 1 + \sum_{a \in \mathcal{A}(I)} \sigma_0^i(I, a) \varphi_{\mathcal{D}}^i(J^i(I, a))$$

$$\leq 1 + [\sum_{a \in \mathcal{A}(I), a \neq a_j, a_k} \sigma_0^i(I, a) \varphi_{\mathcal{D}}^i(J^i(I_r, a))] + [\sigma_0^i(I, a_j) + \sigma_0^i(I, a_k)] \varphi^i(J_{\mathcal{D}}^i(I, a_k))$$

With the last inequality, we can construct a $(\sigma_0^i)'$ with $\varphi_{\mathcal{D}}^i((\sigma_0^i)', I_r) \geq \varphi_{\mathcal{D}}^i(\sigma_0^i, I_r)$ and $(\sigma_0^i)'(I') \neq \sigma_0^i(I')$ only if $I' \in subt(I)$. Formally, we define $(\sigma_0^i)'$ as:

- $(\sigma_0^i)'(I') = \sigma_0^i(I'), I' \notin subt(I)$
- $(\sigma_0^i)'(I') = \sigma^{i,*}(I'), I' \in subt(I), I' \neq I$
- $(\sigma_0^i)'(I, a) = \sigma_0^i(I, a), a \neq a_j, a_k$
- $(\sigma_0^i)'(I, a_j) = 0$
- $(\sigma_0^i)'(I, a_k) = \sigma_0^i(I, a_j) + \sigma_0^i(I, a_k)$

As $\sigma_0^i \in \zeta$, $\varphi_{\mathcal{D}}^i((\sigma_0^i)', I_r) = \varphi_{\mathcal{D}}^i(\sigma_0^i, I_r)$, which means $\sigma_0^i(I, a_j) = 0$, due to $\pi_{\sigma_0^i}^i(I) > 0$. Thus, $\forall \sigma_0^i$ such that $\pi_{\sigma_0^i}^i(I) > 0$, we have $\pi_{\sigma_0^i}^i(J^i(I, a_j)) = 0$, which means $J^i(I, a_j) \notin \mathcal{D}$ (notice that if $\pi_{\sigma_0^i}^i(I) = 0, \pi_{\sigma_0^i}^i(J^i(I, a_j)) = 0$), that is contradict to our assumption $J^i(I, a_j) \in \mathcal{D}$. Thus $\varphi_{\mathcal{D}}^i(J^i(I, a_j)) = \varphi_{\mathcal{D}}^i(J^i(I, a_k)), a_j, a_k \in \mathcal{A}_{\mathcal{D}}(I)$.

Now we can show that $\varphi_{\mathcal{D}}^i(\sigma^i, I) = \varphi_{\mathcal{D}}^i(I), \forall \sigma^i, \forall I$ with mathematical induction.

Let $D$ denotes the depth of $\mathcal{D}$. As we assume $P(pa(I)) = i, \forall I \in Z, P(I_r) = i$ if $D$ is odd and $P(I_r) = -i$ if $D$ is even. We separately discuss them as follows:

- $D = 1$: Obviously $\varphi_{\mathcal{D}}^i(\sigma^i, I_r) = \varphi_{\mathcal{D}}(I_r) = 1$.
- $D$ is even: If $\forall a \in \mathcal{A}_{\mathcal{D}}(I_r)$, $J^i(I_r, a)$ satisfies that $\varphi_{\mathcal{D}}^i(\sigma^i, J^i(I_r, a)) = \varphi_{\mathcal{D}}(J^i(I_r, a))$, then

$$\varphi_{\mathcal{D}}^i(\sigma^i, I_r) = \sum_{a \in \mathcal{A}_{\mathcal{D}}(I)} \varphi_{\mathcal{D}}^i(\sigma^i, J^i(I_r, a))$$

$$= \sum_{a \in \mathcal{A}_{\mathcal{D}}(I)} \varphi_{\mathcal{D}}^i(J^i(I_r, a))$$

$$= \varphi_{\mathcal{D}}^i(I_r)$$

Thus we get $\varphi_{\mathcal{D}}^i(\sigma^i, I_r) = \varphi_{\mathcal{D}}^i(I_r)$.

- $D$ is odd and $D \neq 1$: If $\forall a \in \mathcal{A}_\mathcal{D}(I_r)$, $J^i(I_r, a)$ satisfies that $\varphi^i_\mathcal{D}(\sigma^i, J^i(I_r, a)) = \varphi_\mathcal{D}(J^i(I_r, a))$, then $\forall \sigma^i \in \zeta'$,

$$\varphi^i_\mathcal{D}(\sigma^i, I_r)$$
$$=1 + \sum_{a \in \mathcal{A}_\mathcal{D}(I)} \sigma^i(I_r, a) \varphi^i_\mathcal{D}(\sigma^i, J^i(I_r, a))$$
$$=1 + \sum_{a \in \mathcal{A}_\mathcal{D}(I)} \sigma^i(I_r, a) \varphi^i_\mathcal{D}(J^i(I_r, a))$$
$$=1 + [\sum_{a \in \mathcal{A}_\mathcal{D}(I)} \sigma^i(I_r, a)] \cdot \varphi^i_\mathcal{D}(J^i(I_r, a_j)), \forall a_j \in \mathcal{A}_\mathcal{D}(I) \quad \text{(due to } \varphi^i_\mathcal{D}(I_r, a_j) = \varphi^i_\mathcal{D}(I_r, a_k), \forall a_j, a_k \in \mathcal{A}_\mathcal{D}(I))$$
$$=1 + \varphi^i_\mathcal{D}(J^i(I_r, a_j))$$
$$=\varphi^i_\mathcal{D}(I_r) \quad \text{(as } \forall \sigma^i, \varphi^i_\mathcal{D}(\sigma^i, I_r) = 1 + \varphi^i_\mathcal{D}(J^i(I_r, a)), a \in \mathcal{A}_\mathcal{D}(I) \text{ is independent of } \sigma^i)$$

  We can get $\varphi^i_\mathcal{D}(\sigma^i, I_r) = \varphi^i_\mathcal{D}(I_r)$ as well.

So $\forall \sigma^i, \forall A, \varphi^i_\mathcal{D}(\sigma^i, I_r) = \varphi^i_\mathcal{D}(I_r)$. As $\varphi^i_{\mathcal{D},\sigma^i} = \varphi^i_\mathcal{D}(\sigma_i, I) = \varphi^i_\mathcal{D}(I)$ is independent from $\sigma^i$, we finally prove this lemma. $\square$

As $\varphi^i_{\mathcal{D},\sigma^i}$ is independent of the choice of $\sigma^i$, we will drop the subscript $\sigma^i$, that is, we will use $\varphi^i_\mathcal{D}$ instead of $\varphi^i_{\mathcal{D},\sigma^i}$ in the following proof.

Now we prove Theorem 2.

*Proof.* As we discussed before, it's sufficient to focus on the subtreee $\mathcal{D}$, so all of the terms (e.g. the regret) in the following proof are defined on $\mathcal{D}$, not $\mathcal{I}^i$.

For $\mathcal{D}$, $P(pa(I)) = i, \forall I \in Z$, we use the following procedure to generate $\sigma_t^{-i}$ and $u_t^i$ for each round:

- $u_t(I) \sim \text{Bernoulli}(0.5), \forall I \in Z$.

- $\forall I \in \mathcal{D}, P(I) \neq i$, we define $p_\mathcal{D}(I)$ as:

$$p_\mathcal{D}(I) = [\frac{\varphi^i_\mathcal{D}(J^i(I, a_1))}{\sum_{a \in \mathcal{A}_\mathcal{D}(I)} \varphi^i_\mathcal{D}(J^i(I, a))}, \cdots, \frac{\varphi^i_\mathcal{D}(J^i(I, a_{|\mathcal{A}_\mathcal{D}(I)|}))}{\sum_{a \in \mathcal{A}_\mathcal{D}(I)} \varphi^i_\mathcal{D}(J^i(I, a))}]$$

  Notice that this term only depends on $\mathcal{D}$, so once we determine $\mathcal{D}$, we can immediately get this $p_\mathcal{D}(I)$. Each turn we first sample $a(I)$ from $\text{Multinomial}(1, p_\mathcal{D}(I))$, then let $\sigma_t^{-i}(I) = a(I)$.

In the following proof we denote this generating procedure as $\mathcal{M}$ and use the notation $\mathbb{E}_\mathcal{M}[\cdot]$ as the expectation over this generating procedure.

Let $n_t(I), I \in \mathcal{D}$ denotes the cumulative arriving time player $i$ arrives at $I$ in the first $t$ rounds. With a little abuse of notation, we use the term $R_T^i(I), I \in \mathcal{D}$ to represent the regret of the subtree rooted at $I$ in the first $T$ turns, i.e.

$$R_T^i(I) = \max_{\sigma^i} \sum_{t=1}^T u^i((\sigma^i, \sigma_t^{-i}), I) - u^i((\sigma_t^i, \sigma_t^{-i}), I)), \quad I \in \mathcal{D}$$

We will prove that $\lim_{A \to \infty} \lim_{T \to \infty} \mathbb{E}_\mathcal{M} R_{T,alg}^i / \sqrt{\frac{\varphi^i_\mathcal{D} T \log A}{2}} \geq 1, \forall alg$ with mathematical induction, where $R_{T,alg}^i$ is defined in Theorem 2. Notice that under the assumption that $P(pa(I)) = i, \forall I \in Z$, $P(I_r) = i$ if $D$ is odd while $P(I_r) = -i$ if $D$ is even. In our proof we will discuss them separately.

- $D = 1$: As (Auer et al., 1995; Freund & Schapire, 1997; Cesa-Bianchi & Lugosi, 2006; Dani et al., 2008) have shown, for $K$-arm online linear optimization problem, $\lim_{K\to\infty}\lim_{T\to\infty}\frac{\mathbb{E}_{\mathcal{M}}R^i_{T,alg}}{\sqrt{\frac{T\log K}{2}}} \geq 1, \forall\, alg$, which is consistent to our proposition with $\varphi^i_{\mathcal{D}} = 1$[4].

- $D$ is even: From the definition we can get that $\varphi^i_{\mathcal{D}}(I_r) = \sum_{a\in\mathcal{A}_{\mathcal{D}}(I_r)}\varphi^i_{\mathcal{D}}(J^i(I_r,a))$.

  If $\lim_{A\to\infty}\lim_{T\to\infty}\mathbb{E}_{\mathcal{M}}R^i_T(J^i(I_r,a))/\sqrt{\frac{\varphi^i_Z(J^i(I_r,a))T\log A}{2}} \geq 1, \forall a \in \mathcal{A}_{\mathcal{D}}(I_r), \forall alg$, then

$$
\lim_{A\to\infty}\lim_{T\to\infty}\frac{\mathbb{E}_{\mathcal{M}}R^i_{T,alg}(I_r)}{\sqrt{\frac{\varphi^i_{\mathcal{D}}(I_r)T\log A}{2}}}
$$

$$
= \lim_{A\to\infty}\lim_{T\to\infty}\frac{\sum_{a\in\mathcal{A}_{\mathcal{D}}(I_r)}\mathbb{E}_{\mathcal{M}}R^i_{n_T(J^i(I_r,a)),alg}(J^i(I_r,a))}{\sqrt{\frac{\varphi^i_{\mathcal{D}}(I_r)T\log A}{2}}}
$$

$$
= \lim_{|\mathcal{A}_{\mathcal{D}}(I_r)|\to\infty}\sum_{a\in\mathcal{A}_{\mathcal{D}}(I_r)}\lim_{A\to\infty}\lim_{n_T(J^i(I_r,a))\to\infty}\frac{\mathbb{E}_{\mathcal{M}}R^i_{n_T(J^i(I,a)),alg}(J^i(I_r,a))}{\sqrt{\frac{\varphi^i_{\mathcal{D}}(J^i(I_r,a))n_T(J^i(I_r,a))\log A}{2}}}
$$

$$
\times\sqrt{\frac{\varphi^i_{\mathcal{D}}(J^i(I_r,a))n_T(J^i(I_r,a))}{\varphi^i_{\mathcal{D}}(I_r)T}}
$$

$$
\geq \lim_{A\to\infty}\lim_{T\to\infty}\sum_{a\in\mathcal{A}_{\mathcal{D}}(I_r)}1\times\frac{\varphi^i_{\mathcal{D}}(J^i(I,a))}{\sum_{a\in\mathcal{A}_{\mathcal{D}}(I_r)}\varphi^i_{\mathcal{D}}(J^i(I,a))}
$$

$$
= 1
$$

  In the first equation, we decompose the overall expected regret to the summation of regrets on subtrees. This decomposition can be derived in a similar way to that of Eq. (2) in Appendix C.

  In the second equation we transform the limitation of $T$ into limitation of $n_T(J^i(I^r,a)), \forall a \in \mathcal{A}_{\mathcal{D}}(I_r)$. As the adversary selects action $a_j$ with probability $p_{\mathcal{D}}(I_r)_j > 0$ where $p_{\mathcal{D}}(I_r)_j$ denotes the $j$-th element of $p_{\mathcal{D}}(I_r)$, when $T \to \infty$, the $n_T(J^i(I_r,a)) \to \infty, \forall a \in \mathcal{A}_{\mathcal{D}}(I_r)$ as well.

  The inequality is from induction and $\lim_{T\to\infty}n_T(J^i(I_r,a))/T = p_{\mathcal{D}}(I)_j, a_j \in \mathcal{A}_{\mathcal{D}}(I_r)$ due to the strong law of large numbers.

- $D$ is odd and $D \neq 1$: For convenience, let $R^i_{T,alg,imm}(I_r) = \max_{a\in\mathcal{A}_{\mathcal{D}}(I_r)}\sum_{t=1}^T u_t(\sigma_t|_{I_r\to a}, I_r) - u_t(\sigma_t, I_r)$ denote the immediate regret on the root node for algorithm $alg$. We first show that under our construction $\mathbb{E}_{\mathcal{M}}R^i_{T,alg,imm}(I_r) \geq 0, \forall\, alg$.

---

[4](Cesa-Bianchi & Lugosi, 2006) proves that for K-arm online linear optimization problem, $\sup_{T,K}\max_{u_t}\frac{\mathbb{E}R_{T,alg}}{\sqrt{T\log K/2}} \geq 1$, however the supremum can only get with $K \to \infty$ and $T/K \to \infty$ as they use the property of maximum of infinite normal random variable and central limit theorem (CLT) correspondingly. Here for clarity we equivalently change supremum into limit in our proof.

Notice that in our construction, $\mathbb{E}_{\mathcal{M}} \sum_{t=1}^{T} u_t(\sigma_t|_{I_r \to a}, I_r) = 0.5T, \forall a \in \mathcal{A}_{\mathcal{D}}(I_r)$. Meanwhile, $\mathbb{E}_{\mathcal{M}} \sum_{t=1}^{T} u_t(\sigma_t, I_r) = 0.5T$ as well. Thus,

$$
\begin{aligned}
\mathbb{E}_{\mathcal{M}} \sum_{t=1}^{T} u_t(\sigma_t, I_r) =& \frac{1}{A} \sum_{a \in \mathcal{A}_{\mathcal{D}}(I_r)} \mathbb{E}_{\mathcal{M}} u_t(\sigma_t|_{I_r \to a}, I_r) \\
=& \mathbb{E}_{\mathcal{M}} \frac{1}{A} \sum_{a \in \mathcal{A}_{\mathcal{D}}(I_r)} u_t(\sigma_t|_{I_r \to a}, I_r) \\
\leq& \mathbb{E}_{\mathcal{M}} \max_{a \in \mathcal{A}_{\mathcal{D}}(I_r)} u_t(\sigma_t|_{I_r \to a}, I_r)
\end{aligned}
$$

This inequality is true for all $\sigma_t$ (i.e. $alg$), and we can get

$$
\mathbb{E}_{\mathcal{M}} R_{T,alg,imm}^i(I_r) = \mathbb{E}_{\mathcal{M}} \max_{a \in \mathcal{A}_{\mathcal{D}}(I_r)} \sum_{t=1}^{T} u_t(\sigma_t|_{I_r \to a}, I_r) - u_t(\sigma_t, I_r) \geq 0, \quad \forall \, alg
$$

Then, similar to the case when $D$ is even, we can get:

$$
\begin{aligned}
& \lim_{A \to \infty} \lim_{T \to \infty} \frac{\mathbb{E}_{\mathcal{M}} R_{T,alg}^i(I_r)}{\sqrt{\frac{\varphi_{\mathcal{D}}^i(I_r) T \log A}{2}}} \\
=& \lim_{A \to \infty} \lim_{T \to \infty} \frac{\mathbb{E}_{\mathcal{M}} [R_{T,alg,imm}^i(I_r) + \sum_{a \in \mathcal{A}_{\mathcal{D}}(I_r)} [R_{n_T(J^i(I_r,a)),alg}^i(J^i(I_r,a))]]}{\sqrt{\frac{\varphi_{\mathcal{D}}^i(I_r) T \log A}{2}}} \\
\geq& 0 + \lim_{A \to \infty} \lim_{T \to \infty} \sum_{a \in \mathcal{A}_{\mathcal{D}}(I_r), n_T(J^i(I_r,a)) < \infty} \frac{\mathbb{E}_{\mathcal{M}} R_{n_T(J^i(I_r,a)),alg}^i(J^i(I_r,a))}{\sqrt{\frac{\varphi_{\mathcal{D}}^i(I_r) T \log A}{2}}} \\
& + \lim_{|\mathcal{A}_{\mathcal{D}}(I_r)| \to \infty} \sum_{a \in \mathcal{A}_{\mathcal{D}}(I_r), n_T(J^i(I_r,a)) \to \infty} \lim_{A \to \infty} \lim_{n_T(J^i(I_r,a)) \to \infty,} \frac{\mathbb{E}_{\mathcal{M}} R_{n_T(J^i(I_r,a)),alg}^i(J^i(I_r,a))}{\sqrt{\frac{\varphi_{\mathcal{D}}^i(I_r) T \log A}{2}}} \\
\geq& 0 + 0 + \lim_{A \to \infty} \lim_{T \to \infty} \sum_{a \in \mathcal{A}_{\mathcal{D}}(I_r), n_T(J^i(I_r,a)) \to \infty} \sqrt{\frac{\varphi_{\mathcal{D}}^i(J^i(I_r,a)) n_T(J^i(I_r,a))}{\varphi_{\mathcal{D}}^i(I_r) T}} \\
\geq& \lim_{A \to \infty} \lim_{T \to \infty} \sqrt{\frac{\sum_{a \in \mathcal{A}_{\mathcal{D}}(I_r), n_T(J^i(I_r,a)) \to \infty} \varphi_{\mathcal{D}}^i(J^i(I_r,a)) n_T(J^i(I_r,a))}{\varphi_{\mathcal{D}}^i(I_r) T}} \\
=& \lim_{A \to \infty} \sqrt{\frac{\varphi_{\mathcal{D}}^i(I_r) - 1}{\varphi_{\mathcal{D}}^i(I_r)}} \\
=& 1
\end{aligned}
$$

Similarly, we decompose the overall expected regret to each subtree in the first equation.

Notice that $\forall \, alg$, $\lim_{T \to \infty} \sum_{a \in \mathcal{A}_{\mathcal{D}}(I_r), n_T(J^i(I_r,a)) < \infty} \mathbb{E}_{\mathcal{M}} R_{n_T(J^i(I_r,a)),alg}^i < \infty$, $\lim_{T \to \infty} \sum_{a \in \mathcal{A}_{\mathcal{D}}(I_r), n_T(J^i(I_r,a)) < \infty} \mathbb{E}_{\mathcal{M}} R_{n_T(J^i(I_r,a)),alg}^i / \sqrt{T} = 0$. Thus the second inequality is true. The last inequality is true by $\sum_i \sqrt{a_i} \geq \sqrt{\sum_i a_i}$.

Notice that $\sum_{a \in \mathcal{A}_{\mathcal{D}}(I_r)} \varphi_{\mathcal{D}}^i(J^i(I_r,a)) n_T(J^i(I_r,a)) = (\varphi_{\mathcal{D}}^i(I_r) - 1)T$ (We can see this by $n_T(J^i(I_r,a)) = \sum_{t=1}^{T} \sigma_t^i(I_r,a)$ and $\forall t, \sum_{a \in \mathcal{A}_{\mathcal{D}}(I_r)} \varphi_{\mathcal{D}}^i(J^i(I_r,a)) \sigma_t^i(I_r,a) = \varphi_{\mathcal{D}}^i(I_r) - 1$, which can be simply derived by the definition of $\varphi_{\mathcal{D}}^i$ and $P(I_r) = i$), and $\lim_{T \to \infty} \sum_{a \in \mathcal{A}_{\mathcal{D}}(I_r), n_T(J^i(I_r,a)) < \infty} \frac{\varphi_Z^i(J^i(I_r,a)) n_T(J^i(I_r,a))}{\varphi_Z^i(I_r) T} = 0$. Thus we can get the limitation of $T \to \infty$.

$\square$

Thus, with mathematical induction, we prove that $\lim_{T\to\infty, A\to\infty} \mathbb{E}_{\mathcal{M}} \frac{R_T^i}{\sqrt{\frac{\varphi_{\mathcal{D}}^i T \log A}{2}}} \geq 1$, while with

the definition of $\mathcal{D}$ and Lemma 4, we can get $\varphi_{\mathcal{D}}^i = \varphi^i$, thus we can get the mini-max lower bound in Theorem 2.

## E    THE DETAILS OF IMPLEMENTATION

We now discuss the detailed implementation of Lazy-CFR. To start, we derive Eq. (4). According to Property 1, we have:

$$
\begin{aligned}
m_t(h) &= \sum_{t'=\tau_t(h)+1}^{t} \pi_{\sigma_{t'}}^{-i}(h) \\
&= \sum_{h':h\in subt(h')} \sum_{t'=t_1(h')+1}^{t_2(h')} \pi_{\sigma_{t'}}^{-i}(h) \\
&= \sum_{h':h\in subt(h')} \sum_{t'=t_1(h')+1}^{t_2(h')} \pi_{\sigma_{t'}}^{-i}(h') \pi_{\sigma_{t'}}^{-i}(h', h) \\
&= \sum_{h':h\in subt(h')} \pi_{\sigma_\Gamma}^{-i}(h', h) \hat{\alpha}(h')
\end{aligned}
$$

The first equation is the definition of $m_t$; the second line is according to Prop 1.2; the third line is derived from the definition of $\alpha$ and $\pi_\sigma^{-i}$ and the last line is due to Property 1.3. Similarly, we can write $r_j$ as:

$$
\begin{aligned}
r_j(h, a) &= \sum_{t'=\tau_t(h)+1}^{t} \pi_{\sigma_{t'}}^{-i}(h) u^i((h, a)|\sigma_{t'}) \\
&= \sum_{h':h\in subt(h')} \sum_{t'=t_1(h')+1}^{t_2(h')} \pi_{\sigma_{t'}}^{-i}(h) u^i((h, a)|\sigma_{t'}) \\
&= \alpha(h) + \sum_{h'\neq h:h\in subt(h')} \sum_{t'=t_1(h')+1}^{t_2(h')} \pi_{\sigma_{t'}}^{-i}(h') \pi_{\sigma_{t'}}^{-i}(h', h) u^i((h, a)|\sigma_{t'}) \\
&= \alpha(h) + U((h, a)) \sum_{h'\neq h:h\in subt(h')} \pi_{\sigma_\Gamma}^{-i}(h', h) \hat{\alpha}(h')
\end{aligned}
$$

We now present how to implement of the ideas in 3.2.1.

---

**Algorithm 2** A detailed implementation of Lazy-CFR

---

1: A two-player zero-sum extensive game.
2: Randomly initialize $\Gamma$.
3: $\forall h \in H, i \in \{1, 2\}$, compute the counterfactual reward $U^i(h) = u^i(h|\Gamma), flag^i(h) = -1$.
4: $\forall I \in \mathcal{I}, s(I) = 0, \forall a \in \mathcal{A}(I), r(I, a) = 0$.
5: **while** $t < T$ **do**
6:    **for all** $i \in \{1, 2\}$ **do**
7:       $\alpha^i(h_r) + = 1.0$.
8:       UPDATE1$(I_r^i, i)$ where $I_r^i$ is the root of infoset tree $\mathcal{I}^i$.
9:    **end for**
10:   $\forall i \in \{1, 2\}$ UPDATE2$(h_r, i, t)$ where $h_r$ denotes the root of the history tree.
11: **end while**
12: **RETURN** $\bar{\sigma}$.

---

**Algorithm 3** UPDATE1$(I, i, t)$

---

1: $m(I) = 0$
2: **for all** $h \in I$ **do**
3:    $\theta^i(h) = (\theta^i(pa(h)) + \hat{\alpha}^i(pa(h))) \times \pi_\Gamma^{-i}(pa(h), h)$.
4:    $m(I) + = \alpha^i(h) + \theta^i(h)$
5: **end for**
6: **if** $m(I) \geq 1$ **then**
7:    **if** $P(I) = i$ **then**
8:       **for all** $a \in \mathcal{A}(I)$ **do**
9:          UPDATE1$((I, a), t)$.
10:       **end for**
11:       $\forall h \in I$, updflag$(h, t, i)$.
12:       $\forall a \in \mathcal{A}(I), r(I, a) = 0$.
13:       **for all** $a \in \mathcal{A}(I), h \in I$ **do**
14:          $r(I, a) + = \beta(h, a) + \theta^i(h)U^i((h, a))$.
15:       **end for**
16:       $\Gamma^i(I) =$RM$(r(I))$.
17:    **else**
18:       $\forall a \in \mathcal{A}(I)$, UPDATE1$((I,a),i, t)$.
19:    **end if**
20: **end if**

---

**Algorithm 4** updflag$(h, t, i)$

---

1: **if** $h$ is not the root of history tree and $flag^i(h) \neq t$ **then**
2:    $flag^i(h) = t$.
3:    $updflag(pa(h), t, i)$.
4: **end if**

---

**Algorithm 5** UPDATE2$(h, i, t)$

---

1: Let $i'$ denote the opponent of player $i$.
2: **if** $flag^i(h) = t$ **then**
3:    $\forall a \in \mathcal{A}(h)$ UPDATE2$((h, a), i, t)$.
4:    update $U^1(h)$ and $U^2(h)$.
5:    $\alpha^i(h) = 0$.
6:    $\hat{\alpha}^{i'}(h) = 0$.
7:    $\forall a \in \mathcal{A}(h), \beta^{i'}(h, a) = 0$.
8: **else**
9:    $\alpha^{i'}(h) + = (\theta^{i'}(pa(h)) + \hat{\alpha}^{i'}(pa(h))) \times \pi_{\Gamma'}^{-i'}(pa(h), h)$.
10:    $\hat{\alpha}^{i'}(h) + = (\theta^{i'}(pa(h)) + \hat{\alpha}^{i'}(pa(h))) \times \pi_{\Gamma'}^{-i'}(pa(h), h)$.
11:    $\forall a \in \mathcal{A}(h), \beta^{i'}(h, a) + = U^{i'}(h, a)(\theta^{i'}(pa(h)) + \hat{\alpha}^{i'}(pa(h))) \times \pi_{\Gamma'}^{-i'}(pa(h), h)$.
12: **end if**

---

