# OpenReview forum: "Lazy-CFR: fast and near-optimal regret minimization for extensive games with imperfect information"
_ICLR.cc/2020/Conference — Accept (Poster)_

### Official Review · AnonReviewer2 · 2019-10-20
**Official Blind Review #3**

**Rating:** 8

**Review:**

This paper introduces lazy-CFR, a CFR algorithm that updates only a subset of information sets each round (but notably differs from pruning and Monte Carlo methods). The paper offers a nice review of online linear optimization and its relationship to CFR, making its proposed algorithm easily digestible. The paper establishes convergence guarantees for lazy-CFR and shows experimentally that lazy-CFR+ outperforms CFR+, a formidable baseline, in Leduc poker. As is discussed in the paper, a major drawback of lazy-CFR is that its space complexity is on the order of the number of histories in the game. This will be an important direction for future work.

This is a nice paper. The writing is strong and its main idea is both novel and appears to be effective in practice. I have no major criticisms. What I would most like to see is more extensive experimental results. Does lazy-CFR offer similarly strong results other small imperfect information games (Kuhn poker, liar’s dice, etc.)? Additionally, it would be interesting to see how the how lazy-CFR performs compared 1) against discounted-CFR and 2) when combined with discounted-CFR.

**Experience Assessment:**

I have read many papers in this area.

**Review Assessment: Checking Correctness Of Derivations And Theory:**

I assessed the sensibility of the derivations and theory.

**Review Assessment: Checking Correctness Of Experiments:**

I assessed the sensibility of the experiments.

**Review Assessment: Thoroughness In Paper Reading:**

I read the paper thoroughly.

---

> ### Author Response · Authors · 2019-11-12
> **Response to Reviewer 2**
>
> Thanks for your acknowledgement on our contributions. We have added some additional experiments, including the evaluations on Linear CFR, Lazy-Linear CFR and a larger game. Please see our response to Reviewer 3 and Sec. 6 in the revision for more details.

---

### Official Review · AnonReviewer3 · 2019-10-21
**Official Blind Review #3**

**Rating:** 8

**Review:**

The paper proposes an improvement to Counterfactual Regret Minimization, avoiding traversing the whole tree on each iteration. The idea is not to change the strategy in those infosets, where the reach probability of opponents is low. The strategy in such infosets is only updated once in several iterations, when the sum of reach probabilities over these iterations if higher than the threshold. The straightforward implementation of the idea still has the same running time as CFR. Therefore, the paper presents an efficient implementation, exploiting the structure of the game tree. However, this implementation comes at the cost of additional memory requirements. Overall, the paper proves the theoretical result of about O(sqrt(|I|)/D) times faster than CFR to achieve the same approximation error, while the memory requirements increase by a factor of O(|H|/|I|). Here |I| is the number of infosets, D is the depth of the game tree and |H| is the number of histories.

The idea of eliminating unnecessary computations for infosets with low probability is a valuable contribution. The presented theoretical analysis takes an important place in the series of works refining the regret upper bound of CFR and its variants. The experiment confirms performance of the idea.

That being said, I follow up with some questions/criticism.
1.	Implementation in Section 3.2.1 and Appendix E is rather hard to follow. Is there any intuition on how the segment [\tau_t(h), t] is divided, i.e. what does t_1, t_2 and \tau’(h) mean? Also, clarity could be increased if these variables would be defined before they are used.
2.	How is a segmentation rule for Lazy-RM in OLO designed in such a way, that equation \sum_{i=1}^n \max_a c’_i(a)^2 \approx \sum_{j=1}^T \max_a c_j(a)^2 holds?
3.	Section 3.2: “following step (1)”. (1) is an equation for RM in OLO, probably some other reference was meant.
4.	Recently, Linear Cfr was introduced, which outperforms Cfr+. Thus, citation is needed Brown, Noam and Sandholm, Tuomas “Solving Imperfect-Information Games via Discounted Regret Minimization”. Worth to mention, LazyCfr is straightforwardly compatible with Linear Cfr.
5.	The specified space requirements significantly limit the applicability of the presented Lazy-RM implementation. For example, in state-of-art approaches to solve/resolve No-Limit Holdem (Libratus, DeepStack, Pluribus), either the game tree is too large, making the space requirements unrealistic, or the game tree is small enough for getting a good equilibrium approximation fast even with CFR+.

UPD: score updated

**Experience Assessment:**

I have read many papers in this area.

**Review Assessment: Checking Correctness Of Derivations And Theory:**

I assessed the sensibility of the derivations and theory.

**Review Assessment: Checking Correctness Of Experiments:**

I carefully checked the experiments.

**Review Assessment: Thoroughness In Paper Reading:**

I read the paper thoroughly.

---

> ### Author Response · Authors · 2019-11-12
> **Response to Reviewer 3**
>
> Thank you for acknowledging our contributions as well as giving valuable comments. Below, we address the main concerns.
>
> Q1: Implementation in Sec 3.2.1 and Appendix E:
> Thanks. We improved the clarity and defined the notations before they are used. The intuition is that we divide [\tau_t(h), t] into sub-intervals when there is strategy modified on h’ in the subtree rooted at h. To make the section of implementation more readable, we have added an intuitive discussion on why and how we divide [\tau_t(h), t] into sub-intervals before we elaborate the full details on those data structures. Please see the third paragraph in Sec 3.2.1 for the discussion.
>
> Q2: Explanation for the segmentation rule of Lazy-RM:
> In a general OLO, it is possible that \sum_{i=1}^n \max_a c’_i(a)^2 and \sum_{j=1}^T \max_a c_j(a)^2 have a large gap. However, as shown in Sec. 4, we have proved that the worst case regret bounds for CFR and Lazy-CFR both (approximately) match the worst case regret lower bound well, which implies that \sum_{i=1}^n \max_a c’_i(a)^2\approx\sum_{j=1}^T \max_a c_j(a)^2 holds on the OLOs in the game, at least in the worst case.
>
> Q3: Ambiguity for step (1):
> Thanks for pointing out. Here we refer to the step (1) in Section 3.1. We have revised it.
>
> Q4: Lazy-Linear CFR, MC-Linear CFR with negative regret-pruning and evaluations on a larger game:
> Thanks for the suggestion. We have cited the paper and added the evaluations on Linear CFR (LCFR), and the version with lazy-update (i.e., Lazy-LCFR), MC-LCFR and MC-LCFR with negative regret-pruning (MC-LCFR-P), as shown in Fig.3 in the revision. Furthermore, we have done the empirical evaluations on larger games with about 10^9 histories. Please refer to Fig.3(c) of the revision. On the smaller games, LCFR has the best performance and on the larger game, Lazy-CFR+ outperforms other algorithms significantly. Lazy-CFR+ is faster than MC-LCFR-P and the negative regret-pruning accelerates MC-LCFR with a factor about 3. But Lazy-LCFR does worse than LCFR, this might be because our segmentation rule is designed for OLO with uniform weights but LCFR assigns more weights on later iterations. How to develop efficient lazy update variants for LCFR needs to be explored.
>
> Q5: Space requirement:
> Thanks. Indeed, we agree that memory requirement is a potential limitation, as discussed at the end of Section 4.2. Reducing the space requirement is a key issue in our future improvement. For large game trees, one possible solution is to design better segmentation rules as well as better implementations or use Monte-Carlo estimation so that we do not have to store those data structures on each node. However, it deserves a systematical investigation to develop an algorithm which is as fast as Lazy-CFR with a space complexity comparable with other CFR variants.

---

> > ### Comment · AnonReviewer3 · 2019-11-15
> > **Re: Response to Reviewer 3**
> >
> > Thanks for the clarifications.  Implementation section in the updated version of the paper is easier to follow. The additional experiments add significant value to the results: the methods' relative performance on a more complex FHP game is quite different from the Leduc + the necessity to develop efficient lazy update variants for LCFR.  Space requirements are still a weak spot of the method, however, this is a direction of a future work, rather than a ground for rejecting a paper. I have updated the score.

---

### Official Review · AnonReviewer1 · 2019-10-23
**Official Blind Review #1**

**Rating:** 3

**Review:**

This paper presents a variant of the counterfactual regret minimization (CFR) algorithm called Lazy-CFR for two-player zero-sum imperfect information games. The idea is to postpone visiting an information set as long as the sum of the (opponent) reach probabilities for the information set after the last strategy update is lower than a certain threshold. This pruning strategy allows one to avoid traversing the whole game tree and significantly speeds up the computation of approximate Nash equilibria. The authors provide detailed theoretical analysis of the proposed algorithm and show that the bound of the overall regret of the proposed algorithm is comparable to that of the original CFR. They conduct experiments using Leduc hold’em and show that the proposed approach gives significantly better results than existing CFR-based algorithms. The downside of the proposed algorithm is that it requires a memory of O(|H|) for bookkeeping, which can be very large and makes it difficult for the algorithm to be applied to large games.

I feel ambivalent about this paper. On the one hand, the paper presents a promising idea for significantly speeding up the CFR algorithm with detailed theoretical justifications, but on the other hand, the (potentially huge) requirement for memory makes me unsure about the strength and practical merit of the algorithm compared to other CFR variants.

I am a bit disappointed that the authors did not compare their algorithm with existing pruning methods on the ground that they do not have theoretical guarantee on running time. I think empirical comparison to the state of the art is always useful and should be conducted whenever possible. Would it be difficult to compare the proposed method with, for example, Brown and Sandholm (2019)?

**Experience Assessment:**

I have published one or two papers in this area.

**Review Assessment: Checking Correctness Of Derivations And Theory:**

I did not assess the derivations or theory.

**Review Assessment: Checking Correctness Of Experiments:**

I assessed the sensibility of the experiments.

**Review Assessment: Thoroughness In Paper Reading:**

I read the paper at least twice and used my best judgement in assessing the paper.

---

> ### Author Response · Authors · 2019-11-12
> **Response to reviewer #1**
>
> Thank you for acknowledging our contributions as well as giving the valuable comments. We address the concerns in detail below.
>
> Q1: Memory requirement:
> Indeed, as we discussed at the end of Section 4.2, requiring larger memory is a limitation of our current algorithm. For reasonably sized-games (e.g., FHP), it is still possible to store the data structures in a common storage system (e.g., tens of TB). To further improve, one possible way is to design better segmentation rules as well as better implementations or use Monte-Carlo estimation so that we do not have to store those data structures on each node. However, it is worth of a systematical investigation to develop an algorithm that is as fast as Lazy-CFR with a space complexity comparable with other CFR variants.
>
> Q2: Comparison with the regret-based pruning methods in [1]:
> Thanks for the suggestion. But as mentioned by Noam Brown in his comment, regret-based pruning is orthogonal to our work. That is, we can also apply negative regret-pruning to Lazy-CFR. For fair comparisons, we did not use regret-based pruning. In the future, we’d like to systematically conjoin Lazy-CFR with regret-based pruning, which is expected to further improve.
>
> Finally, we have also added the evaluation against MC-LCFR with negative pruning used in [1], please see Sec. 6 in the revision.
>
> [1] Brown, Noam, and Sandholm, Tuomas. "Superhuman AI for multiplayer poker." Science 365.6456 (2019): 885-890.

---

### Public Comment · ~Noam_Brown2 · 2019-11-07
**Some comments/questions**

I noticed the experiments show MCCFR and CFR as nearly parallel lines. That seems odd to me. I would expect CFR to perform worse initially but eventually overtake MCCFR. Were the experiments done using the alternating updates version of CFR and CFR+?

Not comparing to the regret-based pruning methods seems okay since this technique is orthogonal to those, but the reason that is given seems weird. Yes, it is possible to create games where regret-based pruning and best-response pruning are identical to CFR. But in the worst case, Lazy-CFR is also identical to CFR (e.g., a matrix game). The authors might want to mention Dynamic Thresholding though (Brown et al. AAAI-17) since that uses a somewhat similar (though less aggressive) approach as Lazy-CFR.

The references include 3 arXiv papers, all of which are published in conferences. The authors should cite the conference papers instead.

---

> ### Author Response · Authors · 2019-11-12
> **Response to your comments/questions**
>
> Thanks for your valuable comments. Here are our responses.
>
> Q1: On the experiments:
> First, we did not use alternative update in CFR and CFR+.
>
> Then, as for the performance of MCCFR, we have carefully checked our experiments for multiple times. We found that MCCFR had a good performance might be because: (1) the game trees in our experiments (i.e. Fig.3(a)&(b)) are small; and (2) the number of branches on histories of the chance player is small and there are a large portion of histories with \pi^{-i}(h)=0. In such cases, the variance of external sampling in MCCFR is relatively small, thereby resulting in good performance. In our new experiments on larger games (see Fig.3(c) of the revised paper), CFR indeed finally outperforms MCCFR.
>
> Q2: Discussion on regret-based pruning methods and the worst case of Lazy-CFR:
> Thanks. We have updated our discussion on the pruning-based methods in the revision, Sec. 5.
>
> If the threshold is set to be 1, Lazy-CFR degenerates to CFR if and only if the reach probability contributed by –i is 1 on every decision point of player i. This happens only if there is one decision point for player i. And this is the case of your example.
>
> Q3: Dynamic thresholding:
> Thanks for the suggestion. We have cited it in both Introduction and Related work.
>
> Q4: About arXiv references:
> Thanks. We have updated to cite their conference versions.

---

### Decision · Program_Chairs · 2019-12-19

**Decision:**

Accept (Poster)

**Comment:**

The paper proposed an regret based approach to speed up counterfactural regret minimization. The reviewers find the proposed approach interesting. However, the method require large memory. More experimental comparisons and comparisons pointed out by reviewers and public comments will help improve the paper.